# Ebola virus VP35 hijacks the PKA-CREB1 pathway for replication and pathogenesis by AKIP1 association

Lin Zhu [1,8], Ting Gao[1,8], Yi Huang[2,8], Jing Jin[3], Di Wang[3], Leike Zhang [2], Yanwen Jin[1], Ping Li[1], Yong Hu[1], Yan Wu [2], Hainan Liu[1], Qincai Dong[1], Guangfei Wang[1], Tong Zheng[1], Caiwei Song[1], Yu Bai[3], Xun Zhang[3], Yaoning Liu[3], Weihong Yang[3], Ke Xu[4], Gang Zou[5], Lei Zhao[6], Ruiyuan Cao[6], Wu Zhong [6], Xianzhu Xia[7], Gengfu Xiao [2✉], Xuan Liu [1✉] & Cheng Cao [1✉]

Ebola virus (EBOV), one of the deadliest viruses, is the cause of fatal Ebola virus disease (EVD). The underlying mechanism of viral replication and EBOV-related hemorrhage is not fully understood. Here, we show that EBOV VP35, a cofactor of viral RNA-dependent RNA polymerase, binds human A kinase interacting protein (AKIP1), which consequently activates protein kinase A (PKA) and the PKA-downstream transcription factor CREB1. During EBOV infection, CREB1 is recruited into EBOV ribonucleoprotein complexes in viral inclusion bodies (VIBs) and employed for viral replication. *AKIP1* depletion or PKA-CREB1 inhibition dramatically impairs EBOV replication. Meanwhile, the transcription of several coagulation-related genes, including *THBD* and *SERPINB2*, is substantially upregulated by VP35-dependent CREB1 activation, which may contribute to EBOV-related hemorrhage. The finding that EBOV VP35 hijacks the host PKA-CREB1 signal axis for viral replication and pathogenesis provides novel potential therapeutic approaches against EVD.

[1] Beijing Institute of Biotechnology, Beijing 100039, China. [2] National Biosafety Laboratory, Chinese Academy of Sciences, Wuhan, Hubei 430020, China. [3] Institute of Physical Science and Information Technology, Anhui University, Hefei, Anhui 230601, China. [4] State Key Laboratory of Virology, College of Life Sciences, Wuhan University, Wuhan 430072, China. [5] Insitut Pasteur of Shanghai, Chinese Academy of Sciences, Shanghai 200031, China. [6] National Engineering Research Center for the Emergency Drug, Beijing Institute of Pharmacology and Toxicology, Beijing 100850, China. [7] Changchun Veterinary Research Institute, Chinese Academy of Agricultural Sciences, Changchun 130000, China. [8] These authors contributed equally: Lin Zhu, Ting Gao, Yi Huang. ✉email: xiaogf@wh.iov.cn; liux931932@163.com; caoc@nic.bmi.ac.cn

Ebola virus disease (EVD) is the deadliest infectious disease caused by Ebola virus infection[1]. The largest EVD epidemic from 2013 to 2016 caused more than 28,000 confirmed cases and more than 11,000 deaths[2]. The second-largest outbreak occurring in 2018–2020 in the Equator and North Kivu provinces of the Democratic Republic of the Congo resulted in 3470 infections and 2287 deaths (https://www.afro.who.int/health-topics/ebola-virus-disease). The *Ebolavirus* genus includes six species: *Zaire ebolavirus* (with the virus Ebola virus (EBOV)), *Sudan ebolavirus* (Sudan virus (SUDV)), *Bundibugyo ebolavirus* (Bundibugyo virus (BDBV)), *Tai Forest ebolavirus* (Taï Forest virus (TAFV)), *Reston ebolavirus* (Reston virus (RESTV), nonpathogenic in humans), and the newly described *Bombali ebolavirus* (Bombali virus (BOMV))[3], of which *Zaire ebolavirus* is the most virulent, with a case fatality rate of 40–60%, and is the cause of EVD outbreaks[2,4].

Ebola virus is an enveloped, nonsegmented negative-sense (NNS) RNA virus[5]. The 19 kb viral genome comprises seven genes encoding nucleoprotein (NP), virion protein 35 (VP35), VP40, glycoprotein (GP), VP30, VP24, and large protein (L)[1]. NP, VP35, VP30, and L are associated with the viral RNA genome, forming the nucleocapsid termed the viral ribonucleoprotein complex (RNPs)[5]. Viral inclusion bodies (VIBs) formed in EBOV-infected cells are specialized intracellular compartments serving as sites for EBOV replication and the generation of progeny viral RNPs[6,7]. In VIBs, the EBOV genome is replicated and transcribed by viral polymerase complexes[8]. Among RNPs, VP35 serves as a cofactor of RNA-dependent RNA polymerase (RdRp) and contributes to viral replication by homo-oligomerization through a coiled-coil domain[9], as well as the NTPase and helicase-like activities revealed recently[10]. Several host proteins, such as dynein light chain (LC8), DRBP76, PACT, TRIM6, and Staufen1, are moderately dedicated to viral replication by VP35 association[11–15]. In addition, VP35 is also involved in host innate immune antagonism by inhibiting type I interferon (IFN) production, suppressing RNA silencing, and inhibiting dendritic cell maturation[16–18].

Protein kinase A (PKA) is a well-known sensor of the second messenger cyclic AMP (cAMP), playing fundamental roles in a variety of biological processes[19]. Upon activation, PKA phosphorylates numerous substrates in the cytoplasm and nucleus, including cAMP-responsive element-binding protein 1 (CREB1)[19]. PKA-mediated CREB1 S133 phosphorylation activates the transcription of downstream genes in response to hormonal stimulation in a cAMP-dependent manner. PKA is also involved in hepatitis C virus entry, Zika virus infection, and organelle transport regulation in response to adenovirus infection[20–22]. However, the potential role of PKA in EBOV infection and pathogenesis is still unknown. A kinase interacting protein 1 (AKIP1) is a PKA-interacting protein that binds to the amino terminus of protein kinase A catalytic subunit alpha (PRKACA), which potentiates the translocation of PRKACA into the nucleus[23].

In this study, we find that EBOV VP35 significantly potentiates the activation of the PKA-CREB1 signaling pathway by AKIP1 association, thereby facilitating viral replication and resulting in disordered expression of coagulation-related genes. The crucial roles of the AKIP1-PKA-CREB1 signaling axis in viral replication and pathogenesis may direct the development of antiviral and anticoagulopathy therapies for EVD.

## Results

**EBOV VP35 is associated with host AKIP1.** Yeast two-hybrid screening using EBOV VP35 as bait against the human liver cDNA library suggested that human AKIP1 was a candidate for VP35 association (Supplementary Fig. 1a). To substantiate their interaction in EBOV-targeting hepatic cells, HepG2 cells derived from the human liver were transfected with Flag-VP35 or Flag-vector as control and subjected to anti-Flag immunoprecipitation and immunoblotting with the indicated antibodies. The presence of endogenous AKIP1 in the Flag-VP35, but not the Flag-vector, immunoprecipitates suggested the association of VP35 with AKIP1 (Fig. 1a). Associations of VP35 and AKIP1 carrying Flag- and Myc-tags were also observed via reciprocal pulldown in HEK293 cells (Fig. 1b and Supplementary Fig. 1b). Notably, no association was observed between AKIP1 and VP35 of the human nonpathogenic RESTV (Fig. 1c). Then, the crucial sequence of VP35 responsible for AKIP1 association was determined by VP35 truncation and mutation (Supplementary Fig. 1c). We found that the C terminus of VP35 (199-340 amino acids, VP35C) was required for the VP35-AKIP1 interaction (Supplementary Fig. 1d). Notably, triple alanine replacements of F239/K319/R322, which play an essential role in dsRNA binding and inhibiting host type I IFN responses[11,24], resulted in the abrogation of the VP35-AKIP1 association (Fig. 1d and Supplementary Fig. 1c). Nevertheless, the association of VP35 and AKIP1 was observed without the presence of RNA (Fig. 1e). As a control, the association of G3BP1 and cGAS, which was demonstrated to be RNA dependent[25], was abrogated by RNase treatment (Supplementary Fig. 1e). Moreover, an RNA-independent VP35:AKIP1 association was also observed in trVLPs (transcription and replication-competent virus-like particles[26])-infected cells (Supplementary Fig. 1f). For AKIP1, the C-terminal region (101–210 amino acids), but not the N-terminal region (1–100 amino acids), was involved in VP35 association (Fig. 1f and Supplementary Fig. 1c). Moreover, exogenously expressed GFP-VP35 colocalized with Myc-AKIP1, which suggested that AKIP1 interacted with VP35 in the cell (Fig. 1g). Furthermore, an in situ Duolink proximity ligation assay (PLA) was performed to detect VP35:AKIP1 complexes. Cytoplasmic complexes (the red signals) of endogenous AKIP1 with GFP-VP35 (Supplementary Fig. 2a) but not GFP (Supplementary Fig. 2b) were observed in HepG2 cells. Moreover, as a PKA interaction protein, AKIP1 mediated a detectable association between VP35 and protein kinase A catalytic subunit alpha (PRKACA) in wild-type (WT) but not $AKIP1^{-/-}$ HepG2 cells (Supplementary Fig. 2c, d), which eliminated the possibility that VP35 interacted with PRKACA directly or interacted with AKIP1 and PRKACA simultaneously (Supplementary Fig. 2e, f).

We next investigated the association of VP35 and AKIP1 in EBOV- or trVLPs-infected cells. As expected, viral VP35 colocalized with endogenous AKIP1 in live EBOV-infected HepG2 cells (Fig. 2a and Supplementary Fig. 3a), and obvious viral VP35:AKIP1 complexes were also observed in HepG2 cells infected with EBOV or trVLPs as shown by the PLA assay (Fig. 2b). As a control, no PLA signals were observed in uninfected cells (Supplementary Fig. 3b). Based on these results, EBOV VP35 interacts with AKIP1 in host cells, mostly in VIBs (as indicated by the arrow in Fig. 2b).

**EBOV VP35 activates PKA through AKIP1 interaction.** As an activator of PRKACA, AKIP1 demonstrated a significantly enhanced PRKACA binding capability in the presence of VP35 (Fig. 3a), which was also confirmed by in situ PLA in GFP-VP35-expressing cells compared to the control cells (Supplementary Fig. 4a). Moreover, GFP-VP35 (but not GPP) expression potentiated the nuclear translocation of PRKACA in WT but not $AKIP1^{-/-}$ HepG2 cells (Fig. 3b), which may exclude the direct activation of PRKACA by viral VP35. Consequently, WT VP35 (but not VP35 (FKR/AAA) mutant)-transfected HepG2 cells

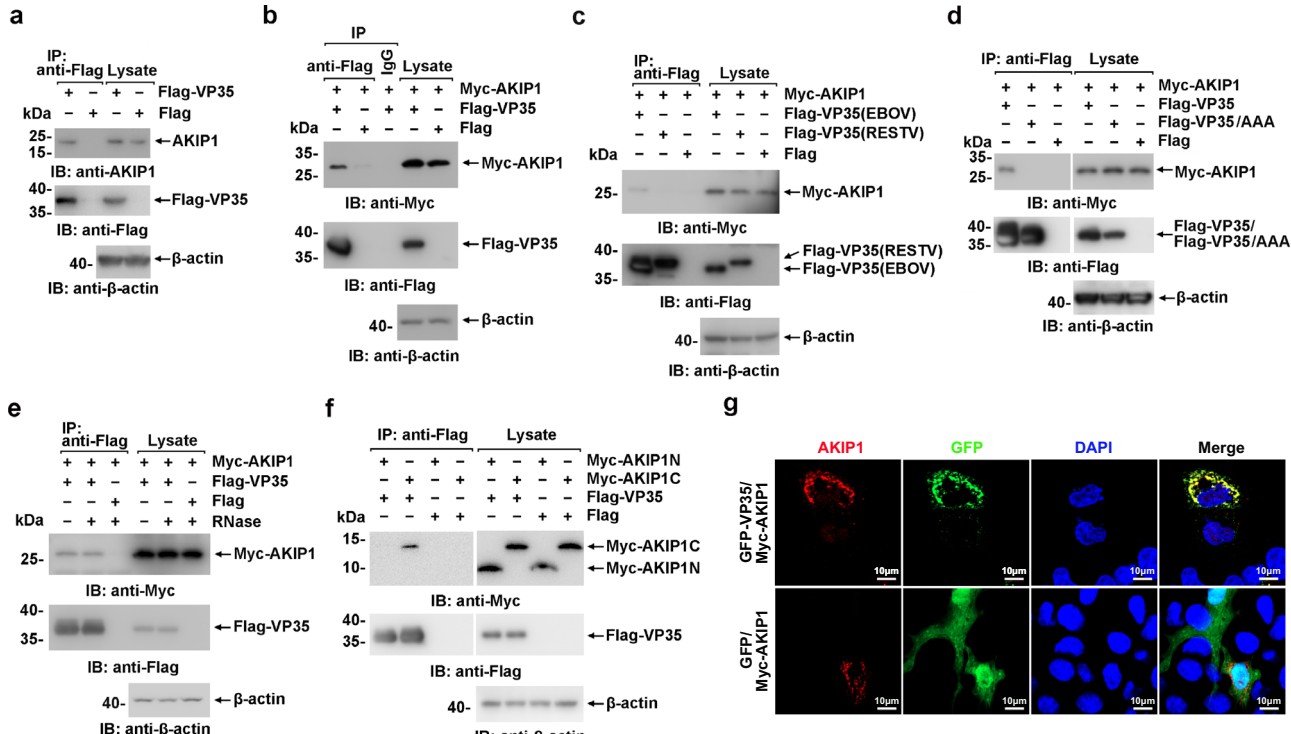

**Fig. 1 The EBOV VP35 associates with AKIP1. a** Lysates of HepG2 cells expressing Flag-VP35 or Flag were subjected to anti-Flag immunoprecipitation and analyzed by immunoblotting. **b–d, f** Lysates of HEK293 cells transfected with the indicated plasmids were subjected to anti-Flag immunoprecipitation and analyzed by immunoblotting. **e** Lysates of HEK293 cells transfected with the indicated plasmids were treated with/without RNase (the mixture of RNase A and RNase T1) and analyzed using immunoprecipitation and immunoblotting. **g** HepG2 cells were cotransfected with GFP-VP35 (or GFP vector) and Myc-AKIP1 and immunostained with an anti-AKIP1 antibody (red). At least three independent repeats were performed in all of the experiments.

showed considerably increased PKA substrate phosphorylation compared with that of the control by anti-pPKA substrate immunoblotting (Fig. 3c). In addition, PRKACA T197 phosphorylation, the hallmark of PKA activation[27], was moderately increased in trVLPs-infected HepG2 cells (Supplementary Fig. 4b). Furthermore, as determined using an in vitro non-radioactive PKA kinase assay, VP35 (but not VP35 (FKR/AAA)) expression potentiated PKA activity in WT (~2.6-fold) but not *AKIP1*-depleted HepG2 cells (Fig. 3d and Supplementary Fig. 4c). Intracellular cAMP concentrations were significantly increased by EBOV infection in wild-type HepG2 cells and, to a lesser extent, in *AKIP1*-depleted HepG2 cells (Fig. 3e). Consequently, VP35 potentiated the phosphorylation of vasodilator-stimulated phosphoprotein (VASP) (Supplementary Fig. 4d) and CREB1 (Fig. 4a), two well-defined PKA substrates[28]. These data collectively suggested that VP35-induced PKA activation is dependent on AKIP1 association. Moreover, at least in the cells we investigated, depletion of *AKIP1* suppressed the transcription of IFN-β and ISG15 (Supplementary Fig. 4e, f). VP35 expression inhibited IFN-β and ISG15 transcription in WT and *AKIP1*−/− HepG2 cells, respectively (Supplementary Fig. 4e, f), suggesting that the interaction of AKIP1 with VP35 at the C terminus did not affect the innate immune antagonistic activity of VP35.

**EBOV VP35 promotes PKA-mediated CREB1 phosphorylation.** We next assessed the effect of VP35 on the activation of CREB1 in detail. CREB1 S133 phosphorylation was significantly potentiated by Ad-VP35 expression, which could be further promoted by FSK, an activator of PRKACA, and suppressed by H89, an inhibitor of PRKACA (Fig. 4a). Moreover, the nuclear accumulation of phosphorylated CREB1 was also induced by Ad-VP35 expression, as well as EBOV or trVLPs infection (Fig. 4b, c).

Consistent with PKA activation, VP35 failed to potentiate CREB1 phosphorylation in *AKIP1* knockdown (Fig. 4d), or *AKIP1*-depleted cells (Fig. 4e), suggesting that viral VP35 potentiates CREB1 phosphorylation through AKIP1. Notably, significantly increased CREB1 S133 phosphorylation was observed in the lung and liver tissues of Ad-VP35-infected mice, but not Ad-null-infected mice (Fig. 4f and Supplementary Fig. 4g, h). These results collectively demonstrated that EBOV VP35 promotes CREB1 phosphorylation in vitro and in vivo in an AKIP1-dependent manner.

**EBOV VP35 hijacks CREB1 into replication complexes in VIBs.** Surprisingly, in addition to facilitating the nuclear accumulation of PRKACA and phosphorylated CREB1, a substantial proportion of PRKACA (Supplementary Fig. 5a) and phosphorylated CREB1 (Fig. 4c and Supplementary Fig. 5b) were colocalized with NP and VP35 in VIBs-like compartments in the cytoplasm when cells were infected with EBOV or trVLPs. In support of this observation, CREB1 was further demonstrated to be colocalized (Fig. 5a, b and Supplementary Fig. 5c, d) and associated (Supplementary Fig. 6) with viral VP35 and NP in VIBs after EBOV or EBOV trVLPs infection. However, this colocalization was abrogated by *AKIP1* depletion, the PKA inhibitor H89, and the CREB1 inhibitor 666-15, resulting in a substantial decrease in the amount and size of VIBs (Fig. 5a, b and Supplementary Fig. 5c, d). In agreement with this observation, CREB1 recruitment into VIBs was similarly detected in cells infected with live EBOV (Fig. 5c, d). These findings suggested that the presence of AKIP1 and PKA-CREB1 activation are indispensable for CREB1 recruitment into VIBs from the cytoplasm.

Lysates of HepG2 cells infected with EBOV trVLPs were fractioned using sucrose density gradient centrifugation to further

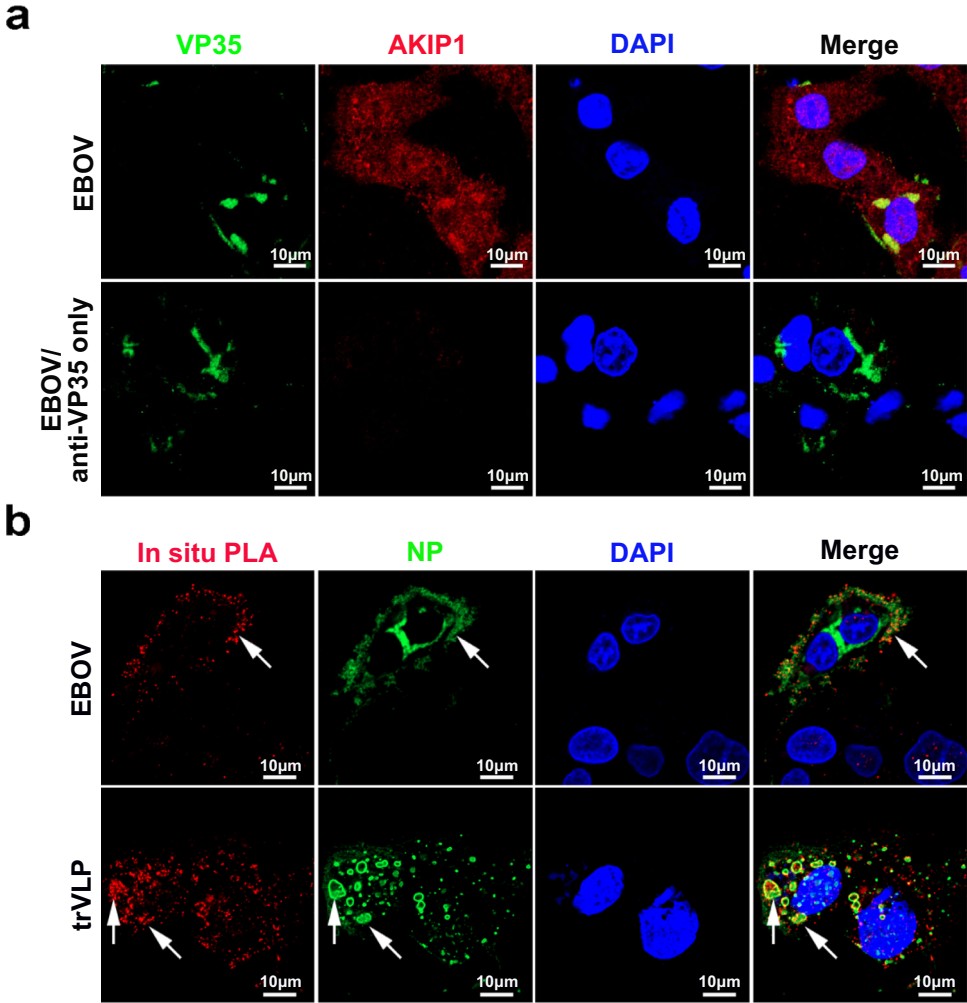

**Fig. 2 EBOV VP35 interacts with AKIP1 in the viral inclusion bodies of HepG2 cells infected with EBOV or trVLPs. a** HepG2 cells infected with Zaire EBOV (strain Mayinga) (MOI = 10) for 72 h were analyzed by immunostaining with anti-VP35 (green) and anti-AKIP1 (red) antibodies (upper panel) or anti-VP35 antibody only (lower panel). **b** HepG2 cells infected with live EBOV (MOI = 10) for 72 h or transfected with the EBOV minigenome (p0) for 48 h were subjected to an in situ PLA assay with anti-VP35 and anti-AKIP1 antibodies (red), and immunostaining with an anti-NP antibody (green). Arrows: VP35-AKIP1 complexes in the viral inclusion bodies of EBOV (upper panel) or trVLPs (lower panel). At least three independent repeats were performed in all of the experiments.

confirm the presence of CREB1 in VIBs, and CREB1 was enriched in the seventh and eighth fractions along with viral VP35 and NP (Fig. 5e). Accordingly, VP35, NP, and viral RNA polymerase L were all present in anti-CREB1 immunoprecipitates prepared from the combined seventh and eighth fractions (Fig. 5f). Depletion of EBOV RNA in the system resulted in slightly slower-migrating complexes (enriched in the seventh fractions) containing CREB1, VP35, and NP (Fig. 5e). Moreover, in EBOV trVLPs-infected HepG2 cells, an RNA-IP assay with an anti-CREB1 antibody showed that, in the presence of AKIP1, CREB1 was more strongly associated with the 3′ leader region of the viral RNA genome containing essential signals required for RNA synthesis[15] but not the 5′ trailer region (Fig. 5g). Importantly, *AKIP1* depletion, CREB1 knockdown, or CREB1 inhibition significantly suppressed the VP35:L association, as well as VIBs formation (Supplementary Fig. 7).

**Active CREB1 is indispensable for EBOV replication.** Activated CREB1 potentates VP35 (an important polymerase cofactor) and L (RNA-dependent RNA polymerase) interactions, suggesting that active CREB1 may be involved in viral replication. EBOV

replication was first evaluated using trVLPs carrying a luciferase reporter (Luc-trVLPs)[26]. Compared with wild-type cells, *AKIP1* depletion resulted in up to a 1300-fold inhibition of Luc-trVLPs replication in two independent cell clones, as indicated by luciferase activity (Fig. 6a, right and Supplementary Fig. 8a), viral genome RNA (vRNA) assays (Fig. 6b), and immunostaining (Supplementary Fig. 8b). AKIP1 knockdown also resulted in significant inhibition of viral replication to a lesser extent (Supplementary Fig. 8c). Overexpression of AKIP1 did not promote EBOV trVLPs activity, possibly because excess AKIP1 may impair the recruitment of AKIP1/CREB1 into the complexes (Supplementary Fig. 8d). As a result of the VP35-AKIP1 association, PKA activation evidently participated in viral replication, as Luc-trVLPs replication was considerably inhibited by ~250-fold in p1 by the PKA inhibitor H89, while it was potentiated ~2.9-fold in p1 by the PKA activator FSK (Fig. 6c and Supplementary Fig. 8a, e). Luc-trVLPs replication was similarly inhibited by H89 in the presence of an anti-IFN-β antibody or JAK1/2 inhibitor (ruxolitinib), which excluded the possibility that PKA promoted the viral replication mainly by attenuating the innate antiviral responses[16,29] (Supplementary Fig. 8f). Furthermore, a much more powerful antiviral effect was observed when the cells were

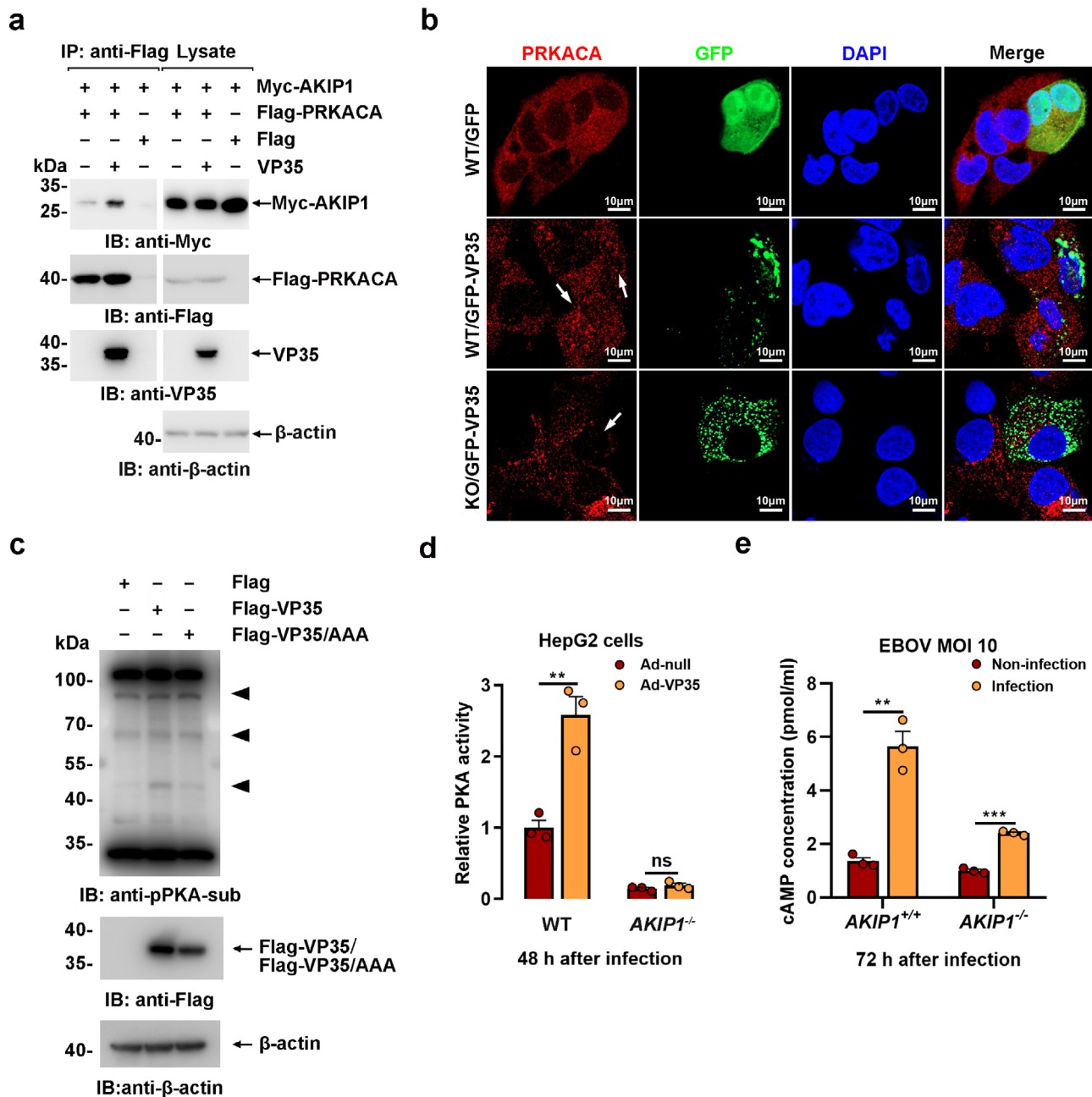

**Fig. 3 EBOV VP35 activates PKA via AKIP1 association. a** Lysates of HEK293 cells cotransfected with the indicated plasmids were incubated with/ without recombinant His-VP35 for 2 h and then subjected to immunoprecipitation and immunoblotting analysis. **b** Wild-type (WT) and *AKIP1* knockout (KO) HepG2 cells transfected with GFP-VP35 or GFP were treated with FSK (25 µM) for 45 min and were subjected to anti-PRKACA immunostaining (red). Arrow: cells expressing GFP-VP35. **c** Lysates of HepG2 cells transfected with Flag-vector, Flag-VP35, or Flag-VP35 mutant plasmids were analyzed by immunoblotting using a PKA substrate phosphorylation antibody. **d** Lysates of WT and *AKIP1*$^{-/-}$ HepG2 cells infected with Ad-VP35 or Ad-null (MOI = 10) were subjected to PKA activity assays. Differences between the two groups were evaluated using a two-sided unpaired Student's *t*-test. Data were presented as mean ± s.e.m. ns not significant; **$P < 0.01$. **e** Concentrations of cAMP were detected in the lysates of WT and *AKIP1*$^{-/-}$ HepG2 cells infected with live EBOV (MOI = 10). Differences between the two groups were evaluated using a two-sided unpaired Student's *t*-test. Data from three independent experiments were analyzed, are presented as the means ± s.e.m. (**$P < 0.01$; ***$P < 0.001$).

treated with the CREB1 inhibitor 666-15 (~6800-fold in p1) (Fig. 6d and Supplementary Fig. 8a, e). The 50% inhibitory concentration (IC50) of 666-15 against EBOV trVLPs was as low as ~50 nM, which was far less than the IC50 of T-705 (~6 µM), a broad-spectrum antiviral candidate that has been clinically used to treat patients with EVD in Sierra Leone[30] (Fig. 6e). Consistent with these findings, the intracellular viral vRNA was substantially inhibited by *AKIP1* depletion (~85-fold), H89 (~19-fold at 10 µM), or 666-15 (~150-fold at 1 µM), as assayed

by qRT-PCR of vRNA (Fig. 6f) or immunostaining (Supplementary Fig. 8g) on the fourth-day post-infection (d.p.i.) with live EBOV. Importantly, live EBOV released into the medium was significantly suppressed by H89 and 666-15, with an IC50 as low as ~500 nM (by 666-15) (Fig. 6e), based on the titer of the viral genomic RNA (Fig. 6g) and the median tissue culture infective dose (TCID$_{50}$) (Fig. 6e, h). These results collectively demonstrated that the AKIP1-PKA-CREB1 axis plays crucial role in EBOV replication.

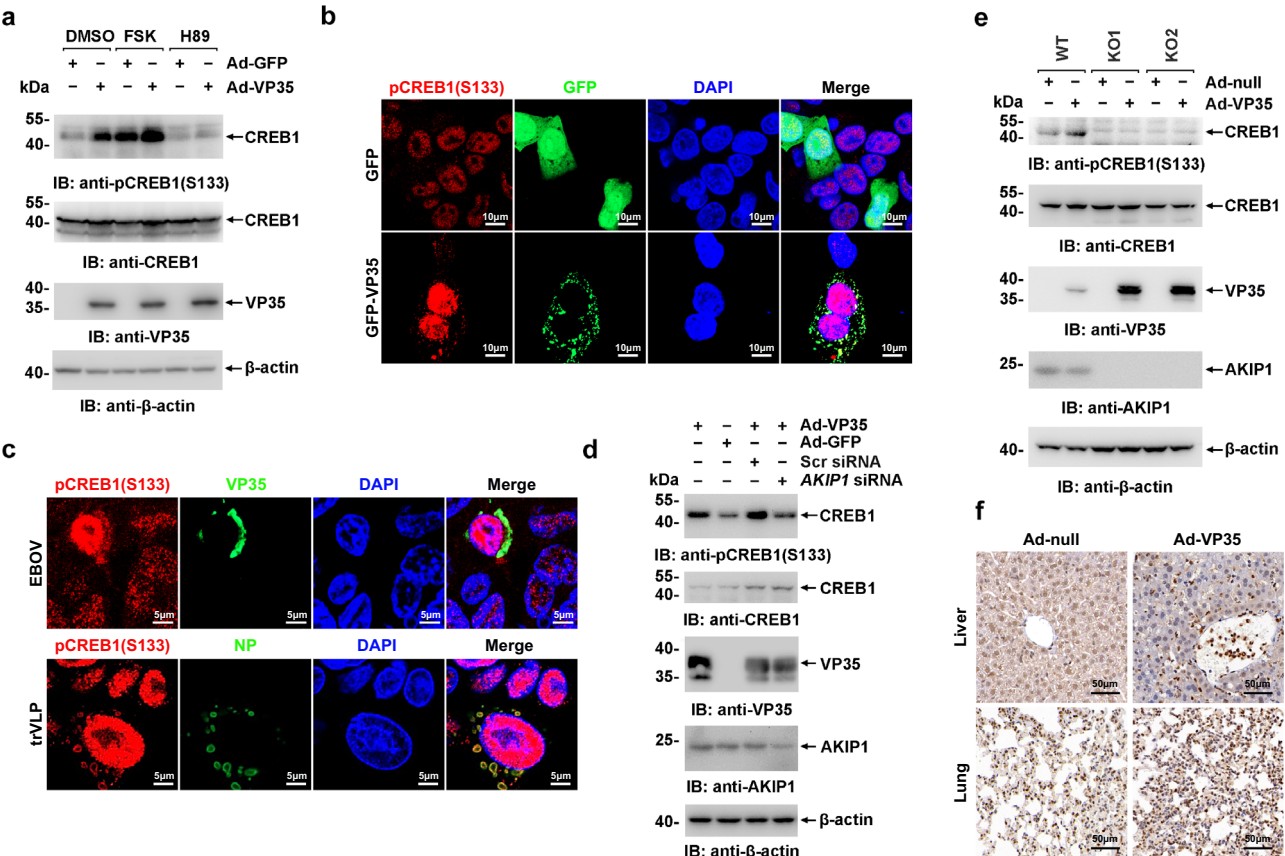

**Fig. 4 EBOV VP35 promotes CREB1 phosphorylation at S133 in vitro and in vivo via AKIP1. a** HepG2 cells infected with Ad-VP35 or Ad-GFP (MOI = 10) for 48 h were treated with 25 μM FSK or 10 μM H89 for 4 h and then analyzed using immunoblotting. **b** HepG2 cells expressing GFP-VP35 or GFP were subjected to immunostaining with an anti-pCREB1(S133) antibody (red). **c** HepG2 cells infected with live EBOV (MOI = 10) for 72 h (upper panel) or transfected with EBOV minigenome (p0) for 48 h (lower panel) were subjected to immunostaining with the indicated antibodies. **d** HepG2 cells transfected with the AKIP1 siRNA or scrambled (Scr) siRNA were infected with Ad-VP35 or Ad-GFP (MOI = 10) for 36 h. Then, lysates were analyzed using immunoblotting. **e** Lysates of WT and *AKIP1*$^{-/-}$ (two independent clones, KO1 and KO2) HepG2 cells infected with Ad-VP35 or Ad-GFP (MOI = 10) were analyzed using immunoblotting. **f** C57BL/6 N mice were intravenously injected with Ad-VP35 or Ad-null (2 × 10$^9$ PFU) twice at an interval of 24 h. Three days after the first infection, the liver (upper panel) and lung (lower panel) tissues were analyzed using immunohistochemical staining with an anti-pCREB1 (S133) antibody. At least two independent replicates were performed in all experiments.

**EBOV VP35 modulates the transcription of CREB1-regulated coagulation-related genes.** Consistent with the findings that VP35 induces CREB1 phosphorylation and nuclear translocation, CREB1-directed transcription was significantly promoted by VP35 expression and further by PKA activation in the luciferase reporter system harboring the CREB1-binding element (Fig. 7a). Then, the transcriptome of HepG2 cells transfected with Flag-VP35 or Flag-vector was investigated using a microarray[31]. Differentially expressed genes (DEGs) were analyzed (Supplementary Fig. 9a), and a panel of coagulation-related genes was found to be significantly enriched (Supplementary Fig. 9b, c), including *THBD*, a biomarker of EBOV-induced hemorrhage and death[32], and *SERPINB2*, a potent inhibitor of urokinase-type plasminogen activator, thrombin and other proteases[33]. The mRNA levels were further quantified by reverse transcription qRT-PCR. Consistent with the results, the mRNA levels of both *THBD* and *SERPINB2* were significantly upregulated by VP35 in HUVECs (Fig. 7b) and HepG2 (Supplementary Fig. 9d) cells. Changes were almost completely abolished by the PKA inhibitor, which suggested that VP35 regulates the gene transcription via the PKA pathway (Fig. 7b and Supplementary Fig. 9d). Accordingly, the protein levels of thrombomodulin (TM, *THBD*-encoded protein) and SerpinB2 were increased by VP35 expression (Supplementary Fig. 9e, f). In agreement, live EBOV infection resulted in a

significant upregulation of *THBD* and *SERPINB2* transcription, which was compromised by H89 or 666-15 treatment, as well as *AKIP1* depletion (Fig. 7c). These results collectively indicate that VP35 may cause coagulation disorder by modifying TM and SerpinB2 expression through the AKIP1-PKA-CREB1 signaling axis.

Furthermore, because no laboratory was permitted to work with animal infection of EBOV in China, Ad-VP35 was employed to assess the effect of EBOV VP35 protein on coagulation. As expected, TM levels were significantly increased in WT mice but not in *Akip1*$^{-/-}$ littermates by Ad-VP35 (not by Ad-null) infection (Fig. 7d and Supplementary Fig. 9g). Ad-VP35 infection also resulted in a prolonged prothrombin time (PT), a reduced fibrinogen (FIB) level, and a subsequently prolonged tail bleeding time in the WT but not *Akip1*$^{-/-}$ mice exposed to Ad-VP35, regardless of challenge with or without LPS, a well-known activator of coagulation[34], changes that were nearly completely rescued by the CREB1 inhibitor 666-15 (Fig. 7e and Supplementary Fig. 9h). Consistent with these findings, Ad-VP35-infected WT mice exhibited significantly higher mortality rates than *Akip1* null mice (77.8 vs. 11.1%) when challenged with LPS (Fig. 7f), and the VP35 potentiated mortality of infected WT mice was completely rescued by the CREB1 inhibitor 666-15 (Fig. 7f). Taken together, these results demonstrated that VP35 causes

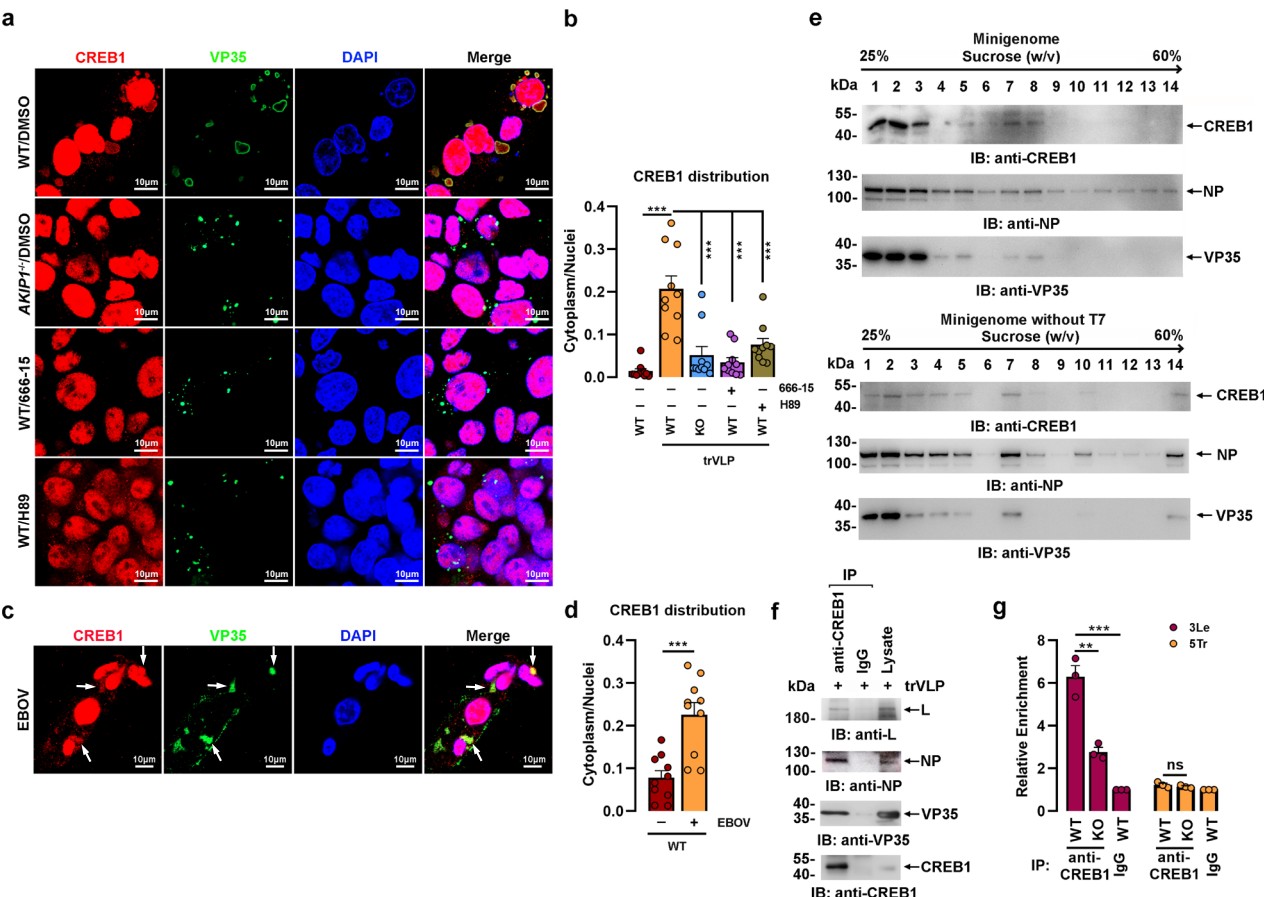

**Fig. 5 CREB1 is recruited to viral inclusion bodies upon trVLPs or EBOV infection. a, b** WT and *AKIP1⁻/⁻* (KO) HepG2 cells were transfected with EBOV minigenome (p0) with or without 1 µM 666-15 or 10 µM H89 for 48 h and then immunostained with anti-VP35 (green) and anti-CREB1 (red) antibodies (**a**). The cytoplasmic/nuclear distribution of CREB1 in (**a**) was analyzed by ImageJ software (**b**). Differences between the two groups were evaluated using a two-sided unpaired Student's *t*-test. The ratio of CREB1 distribution in at least ten cells from two independent assays is presented as the mean ± s.e.m. (*n* = 10; ***P < 0.001). **c, d** HepG2 cells infected with live EBOV (MOI = 10) for 72 h were immunostained with anti-CREB1 (red) and anti-VP35 (green) antibodies. Arrows: CREB1 in VIBs. The ratio of cytoplasm/nuclei distributed CREB1 in (**c**) was analyzed by ImageJ software (**d**). The ratio of CREB1 distribution in at least ten cells from two independent assays is presented as the mean ± s.e.m. (*n* = 10; ***P < 0.001). Differences between the two groups were evaluated using a two-sided unpaired Student's *t*-test. **e** Lysates of HepG2 cells transfected with the EBOV minigenome (p0) in the presence (upper panel) or absence (lower panel) of the T7 RNA polymerase expression plasmid pCAGGS-T7 were separated on a 25 to 60% (W/V) sucrose gradient. Fractions were collected and analyzed by immunoblotting. **f** The seventh and eighth fractions from (**e**, upper panel) were combined and subjected to immunoprecipitation and immunoblotting analysis. **g** WT and *AKIP1⁻/⁻* (KO) HepG2 cells were transfected with the EBOV minigenome (p0). Cell lysates were subjected to anti-CREB1 (or IgG as a control) immunoprecipitation, and the coprecipitated viral RNA corresponding to 3Le or 5Tr was quantified by qRT-PCR. Differences between the two groups were evaluated using a two-sided unpaired Student's *t*-test. The mean ± s.e.m. from three independent assays is presented (ns not significant; **P < 0.01; ***P < 0.001).

coagulation disorder in mice through the AKIP1-PKA-CREB1 pathway. Although other pathways regulated by CREB1 could not be excluded, VP35-induced coagulation disorder may play an important role in LPS-mediated mouse mortality.

In addition to coagulation-related pathways, several other pathways, including the PI3K-Akt signaling pathway, were enriched in the microarray assay. Akt phosphorylation at S473 and T308 was not regulated by VP35 expression (Supplementary Fig. 10), which excludes the possibility that CREB1 activation was mediated by Akt as the activator of CREB1[35].

## Discussion

Accumulating studies of EBOV pathogenesis have revealed a panel of human cellular proteins that participate in or regulate viral replication by interacting with viral proteins[12,36–38] or cis-elements in the viral genome[15]. In this study, PKA was activated by EBOV VP35 through an interaction with AKIP1 that resulted

in the accumulation of PRKACA and increased CREB1 phosphorylation in the nucleus, as expected (Fig. 4b, c). Importantly, the cAMP-responsive transcription factor CREB1 is hijacked into viral replication complexes by EBOV VP35. This process depends on the interaction between viral VP35 and host AKIP1, which activates PKA. VP35, AKIP1, PKA, and CREB1 were observed in the same fraction in Fig. 5e and colocalized in VIBs. Based on these results, PKA could also phosphorylate CREB1 in the cytoplasm, even in the VIBs compartment.

We also observed that most VP35 and NP were recruited into the VIBs compartment other than in cytosol upon trVLPs infection (Fig. 5a and Supplementary Figs. 5c, 6a, 7). In accordance with viral RNA level, VP35 and NP levels in the VIBs were significantly downregulated by AKIP1 depletion, CREB1 knockdown, or 666-15 treatment (Fig. 5a and Supplementary Figs. 5c, 6a, 7), which suggested that VP35 and NP level in VIBs may be used as a marker for virus replication since it might be much more stable than free VP35 in the cytoplasm.

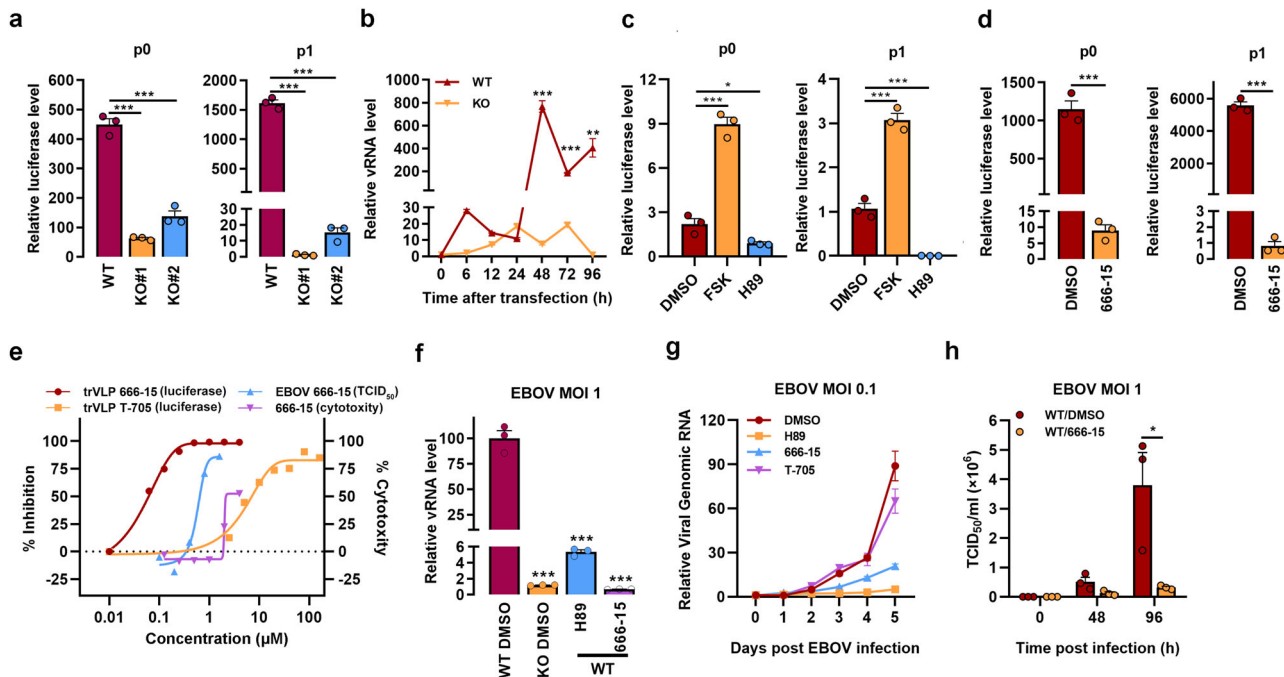

**Fig. 6 AKIP1-PKA-CREB1 potentiates trVLPs and EBOV replication. a, b** WT and *AKIP1*⁻/⁻ (KO) HepG2 cells were transfected with EBOV minigenome (p0 and p1) for 96 h. Cells were lysed and the amounts of trVLPs in the cells were determined by luciferase activity assay (**a**). vRNA was quantified at the indicated times using qRT-PCR (**b**). Differences between the two groups were evaluated using the two-sided unpaired Student's *t*-test. Data were presented as mean ± s.e.m. (***P* < 0.01; ****P* < 0.001). **c, d** HepG2 cells were transfected with the EBOV minigenome (p0 and p1) and treated with FSK (25 μM), H89 (10 μM), or 666-15 (1 μM) for 48 h. The amounts of trVLPs were determined by luciferase activity assay. Differences between the two groups were evaluated using the two-sided unpaired Student's *t*-test. Data are presented as mean ± s.e.m. (**P* < 0.05; ****P* < 0.001). **e** HepG2 cells were transfected with the EBOV minigenome (p1) for 24 h or infected with live EBOV (MOI = 0.1). Then, the cells were treated with the indicated concentration of 666-15 or T-705 as a control for 24 h (for trVLP) and 96 h (for live EBOV infection). The amount of trVLPs was determined by luciferase activity assay, the live EBOV in the supernatant was titered by TCID$_{50}$ assay and the cytotoxicity of 666-15 toward HepG2 cells was determined by performing CCK-8 assay. **f** WT and *AKIP1*⁻/⁻ HepG2 cells infected with live EBOV (MOI = 1) were treated with 10 μM H89 or 1 μM 666-15 for 96 h. vRNA levels in the cells were quantified by qRT-PCR. Differences between the two groups were evaluated using the two-sided unpaired Student's *t*-test. Data were presented as mean ± s.e.m. ****P* < 0.001. **g** HepG2 cells infected with live EBOV (MOI = 0.1) were treated with 10 μM H89, 1 μM 666-15, or 50 μM T-705. The viruses in the supernatants of cell culture were quantified by reverse transcription and qRT-PCR. **h** HepG2 cells infected with live EBOV (MOI = 1) were treated with 1 μM 666-15. The cell culture supernatants were collected at the indicated time points, and the viral titers were quantified as TCID$_{50}$ with plaque assay. Differences between the two groups were evaluated using the two-sided unpaired Student's *t*-test. The data from three independent experiments are presented as the means ± s.e.m. (**P* < 0.05).

As a control, in the cells infected with non-replicative Ad-VP35, or transfected with Flag-VP35 plasmid (Supplementary Fig. 1c), AKIP1 showed little if any effect on VP35 expression in *AKIP1* knockdown cells (Fig. 4d), or even potentiated VP35 expression in *AKIP1* knockout cells (Fig. 4e). These results further demonstrated that AKIP1 regulated viral replication through VP35-induced PKA-CREB1 activation but not a plasmid-driven expression of VP35.

Live EBOV or trVLPs infection resulted in an obvious accumulation of CREB1 in VIBs coexisting with VP35, NP, L, and 3′ leader region of viral genomic RNA, which is essential for viral RNA replication (Fig. 5a, c, f, g). The interaction between RNA polymerase L and its cofactor VP35 was increased by activated CREB1 in VIBs, which represents another possible mechanism for potentiated viral gene transcription and replication. CREB1 promotes mRNA synthesis in cells by interacting with RNA polymerase II through the glutamine-rich domain (Q2)[19] and TBP-associated factor 4 (TAF4)[39], but further investigations are needed to determine whether CREB1 directly promotes RdRp (L protein)-mediated transcription.

The pathogenesis of lethal hemorrhage caused by EBOV has not been well understood until now. The secreted glycoprotein (sGP) of EBOV is considered a virulence factor responsible for vascular dysregulation and hemorrhage by inducing endothelial modifications and lymphocyte adhesion, thereby destroying the barrier function of endothelial cells[40]. However, a recent study also suggested that sGP may not play a key role in vascular dysregulation during EBOV infection[41]. Here, we showed that viral VP35-mediated PKA activation resulted in the phosphorylation of a number of PKA substrates, such as CREB1 presented in this work and VASP (Supplementary Fig. 4d), a well-defined PKA substrate involved in platelet activation and aggregation[28]. Importantly, EBOV infection resulted in the overexpression of *THBD* (encoding TM) and *SERPINB2* (Fig. 7c). TM is ubiquitously expressed on the vascular endothelium, binds to thrombin, forms the TM-thrombin complex, and acts as an anticoagulant. In addition, the thrombin-TM complex activates protein C to produce APC, which inactivates coagulation factors (F) Va and VIIIa in the presence of protein S, thereby inhibiting further thrombin formation[42]. The TM/APC system is considered a guardian of blood coagulation and vascular integrity[42]. Sustained overexpression of TM can result in the exhaustion of thrombin. In fact, TM levels were significantly increased and considered a biomarker for fatal outcomes and hemorrhage in patients with EVD[32]. SerpinB2, as an inhibitor of urokinase-type plasminogen activator, is commonly considered an inhibitor of fibrinolysis. However, a recent study by Schroder et al. indicated that mice with *SERPINB2* deficiency showed significant reductions in tail

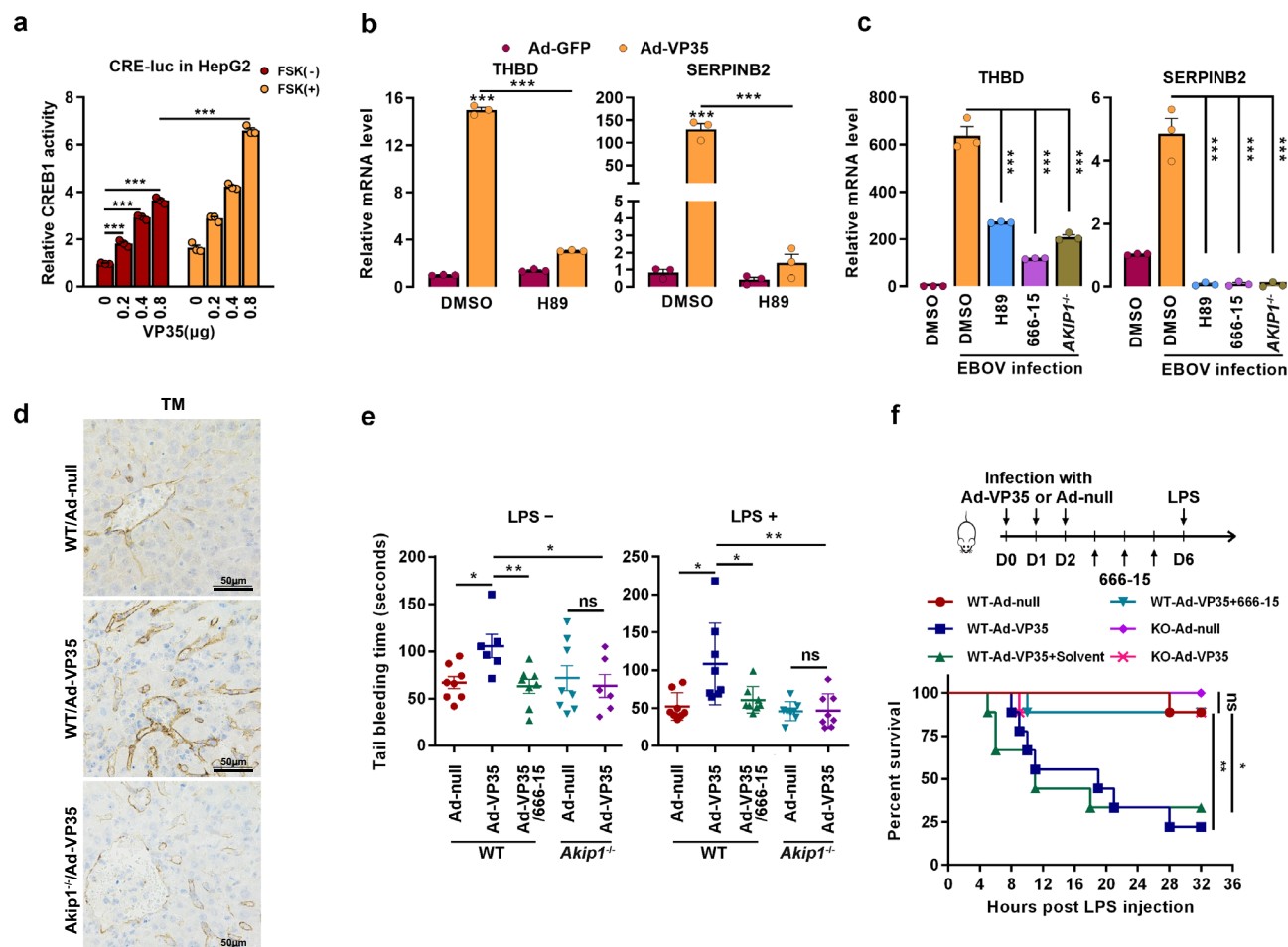

**Fig. 7 EBOV VP35 promotes the transcription of CREB1-directed coagulation-related genes. a** HepG2 cells cotransfected with pGL-CRE-Luc, pRL-TK, and the indicated amounts of Flag-VP35 were treated with or without 25 μM FSK for 4 h. The luciferase activity of the cell lysates was analyzed. Differences between the two groups were evaluated using a two-sided unpaired Student's *t*-test. The mean ± s.e.m. from three independent assays is presented (*n* = 3; ***P* < 0.001). **b**, **c** HUVECs (**b**) and WT or *AKIP1*$^{-/-}$ HepG2 cells (**c**) infected with Ad-VP35 (MOI = 10) (**b**) or live EBOV (MOI = 1) (**c**) for 48 h were treated with or without 10 μM H89 or 1 μM 666-15 for another 24 h. *THBD* and *SERPINB2* mRNA levels were determined by qRT-PCR. Differences between the two groups were evaluated using the two-sided unpaired Student's *t*-test. Data were presented as mean ± s.e.m. (*n* = 3; ****P* < 0.001). **d** WT or *Akip1*$^{-/-}$ mice were intravenously injected with Ad-VP35 or Ad-null (2 × 10$^9$ PFU) twice at an interval of 24 h. Six days post the first infection, the liver tissues were analyzed by immunohistochemistry staining with an anti-Thrombomodulin (TM) antibody. **e**, **f** Mice were infected with Ad-VP35 or Ad-null (3 × 10$^9$ PFU), treated with 666-15 (2 mg/kg) or solvent and then challenged with or without LPS. The tail bleeding time was determined (*n* = 8) (**e**), and a mouse survival curve is shown in (**f**) (*n* = 9). Differences between the two groups were evaluated using the two-sided unpaired Student's *t*-test (**e**). Survival curves were analyzed by log-rank test (**f**). All data from two independent experiments are presented as the means ± s.e.m. (ns not significant; **P* < 0.05; ***P* < 0.01).

bleeding times due to dysregulated platelet activation[33]. Consistent with *THBD* and *SERPINB2* upregulation, Ad-VP35-infected mice displayed a prolonged tail bleeding time (Fig. 7e) and prothrombin time (PT) and a reduced FIB level (Supplementary Fig. 9h). Mice infected with a mouse-adapted strain of EBOV (MA-EBOV) also showed a prolonged thrombin time (TT), PT, and activated partial thromboplastin time (aPTT), as well as decreased FIB levels[43]. Our findings reveal the potential mechanisms of disordered coagulation and extensive microvascular thrombosis induced by viral VP35.

The finding that the AKIP1-PKA-CREB1 signaling axis is hijacked by VP35 and employed for EBOV replication and virus-induced coagulopathy provides new potential targets for the therapy of EBOV-related disease (Fig. 8). Notably, 666-15, a small-molecule CREB1 inhibitor with an IC50 value of ~81 nM that is well tolerated by mice with acceptable pharmacokinetics[44,45], showed promising efficacy in suppressing viral replication (up to

6800-fold) and virus-induced overexpression of TM and SerpinB2. Similar inhibitory efficacy was also observed with the PKA inhibitor H89, although to a lesser extent. Notably, *AKIP1* depletion or CREB1 suppression exerted a smaller effect on live EBOV proliferation than trVLPs (Fig. 6e–h). Low levels of 666-15 (~0.2 μM) somehow boast (but are not statistically significant) the proliferation of live EBOV (Fig. 6e). Compromised IFN-β and ISG15 transcription in *AKIP1* knockout cells (Supplementary Fig. 4e, f) may somehow antagonize the replication inhibitory effect of the CREB1 inhibitor. In addition, downregulation of VP35 levels by AKIP1 expression (Fig. 4e) may also compromise the effect of VP35-induced AKIP1-PKA-CREB1 activation and function as a negative feedback loop. A more powerful inhibition of *AKIP1* depletion or CREB1 suppression of trVLPs replication may be explained by the redundant exogenous VP35 supply in the system. In summary, our findings provide a novel approach for the development of therapeutics against EVD.

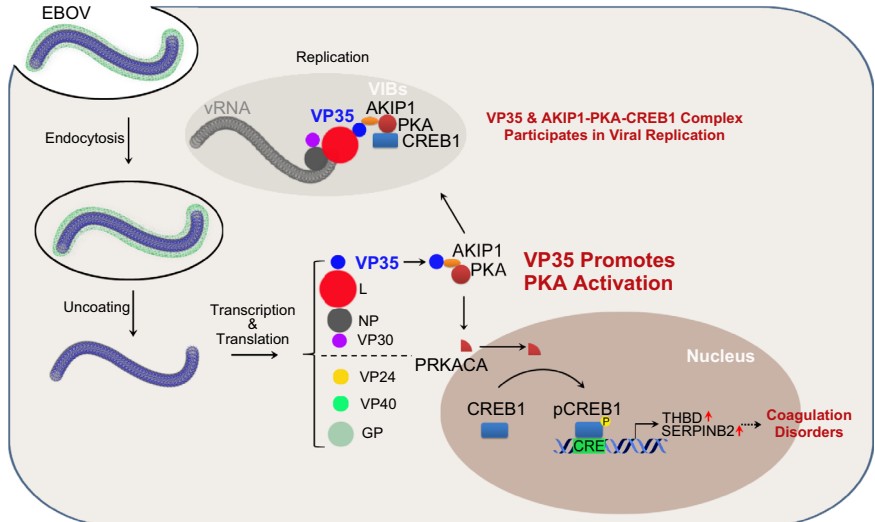

**Fig. 8 Graphical view of the mechanism by which the AKIP1-PKA-CREB1 axis regulated viral replication and coagulation disorders through the VP35:AKIP1 interaction.** EBOV VP35 binds AKIP1, and consequently activates PKA and CREB1. Activated CREB1 is partially recruited into EBOV virus inclusion bodies and potentiates viral replication. And, activated CREB1 translocates into nucleus and promotes the transcription of several coagulation-related genes including *THBD* and *SERPINB2*, which may contribute to EBOV-related hemorrhage.

## Methods

**Cell lines and transfections**. HEK293 and Vero E6 cells were grown in Dulbecco's modified Eagle medium (DMEM, GIBCO). HepG2 cells were grown in Minimum Essential Medium (MEM, GIBCO) supplemented with a 1% Nonessential Amino Acid Solution (NEAA, GIBCO). All media were supplemented with 10% heat-inactivated fetal bovine serum (GIBCO), 2 mM L-glutamine, 100 units/ml penicillin and 100 units/ml streptomycin, and cells were grown at 37 °C under an atmosphere with 5% $CO_2$. Primary human umbilical vein endothelial cells (HUVECs) were grown in EGM-2 medium (Lonza) supplemented with 2% FBS, 0.04% hydrocortisone, 0.4% hEGF-B, 0.1% VEGF, 0.1% R3-IGF-1, 0.1% ascorbic acid, 0.1% hEGF, 0.1% GA-1000, and 0.1% heparin. Cells were treated with forskolin (FSK, Selleck), H89 (Selleck), 666-15 (MCE), T-705 (Selleck), ruxolitinib (Selleck), and lipopolysaccharide (LPS, Sigma), as noted in the text. The cytotoxicity of the drugs used in this study was tested by performing the CCK-8 assay (Dojindo) and trypan blue staining (Invitrogen). Transient transfection was performed with Lipofectamine 3000 (Invitrogen) according to the manufacturer's instructions.

**Mice**. C57BL/6 N AKIP1 knockout ($Akip1^{-/-}$) mice were generated by CRISPR-Cas9-mediated gene targeting by Cyagen Biosciences Inc., Suzhou. Briefly, Cas9 and AKIP1 gRNAs targeting exon 4 (TGATCTGACTCCATCAGGCGAGG and ATACTACTTGTCTATGCCAGAGG) were coinjected into fertilized eggs. The $Akip1^{-/-}$ genotype was confirmed by PCR followed by sequence analysis using primers (mouse $Akip1$-F: 5′- GTTCTCTCCCCAGCTTCTCAGTC -3′; mouse $Akip1$-R: 5′- GCACCCATGTAGTTGAAAATAAAGC -3′), and 28 bp from exon 4 were confirmed to be deleted. WT C57BL/6 N mice (except littermates of $Akip1^{-/-}$ C57BL/6 N mice) and BALB/c mice were purchased from Beijing Vital River Laboratory Animal Technology Co., Ltd. Ten-week-old mice were used in the experiment otherwise indicated. Mice were maintained in a pathogen-free, temperature-controlled, and 12 h light and dark cycle environment with the temperature maintained at 21–23 °C, relative humidity at 50–60%, and free access to food and water at the Academy of Military Medical Sciences. All animal experiments were performed with the approval of the ethics committee at the Beijing Institute of Biotechnology, Beijing, China, and conformed to the relevant regulatory standards. All animal studies were completed in the experimental animal center of the Academy of Military Medical Sciences, China (license number: SCXK-(Army) 2007-004, licensed by the Ministry of Science and Technology of China).

**Vectors and viruses**. Flag-tagged VP35, PRKACA (the catalytic subunit of PKA), cGAS, and VP35 mutants (F239A/K319A/R322A) were constructed by cloning the gene fragments into a pcDNA3.0-based Flag-vector (Invitrogen). Myc-tagged AKIP1 and mutants were constructed by inserting the gene fragments into the pCMV-Myc-vector (Clontech). GFP-tagged VP35 and G3BP1 were expressed by cloning the genes into pEGFP-C1 (Clontech, Takara Bio). All the constructs were validated by Sanger DNA sequencing.

Recombinant adenoviruses (type 5 adenoviruses with deletion of E1a and E3a genes) expressing VP35 (Ad-VP35) were obtained from Beijing BAC Biological Technologies. Analogous adenoviruses expressing GFP (Ad-GFP) or null (Ad-null) were used as controls. Sendai virus (SeV) was amplified in 9- to 11-day embryonated specific-pathogen-free (SPF) eggs. Live EBOV (Mayinga strain) is

preserved by the BSL-4 Lab at the Wuhan Institute of Virology, Chinese Academy of Sciences.

**Yeast two-hybrid screening**. To perform yeast two-hybrid screening, AKIP1 was inserted into the bait pGBKT7 vector for expression as a fusion protein with the Gal4 DNA binding domain (Gal4-BD). This plasmid was used to transform the AH109 yeast strain, and Gal4-BD-fused VP35 was used as bait for the screening of the human liver cDNA library. The liver cDNA library was inserted into the pGADT7 vector (Clontech) for expression as fusions with the Gal4 activation domain (Gal4-AD) and was maintained in the Y187 strain of yeast. Transformed AH109 and Y187 yeast cells were mixed together for mating. Positive clones were selected on synthetic dropout medium lacking 4 nutrients (Leu/Trp/Ade/His). The blue colonies were selected, and the positive results were confirmed by repeating assays. cDNA plasmids isolated from positive colonies were introduced into *Escherichia coli* DH5α and sequenced. The sequences were analyzed with the BLAST program in NCBI.

**Immunoprecipitation and immunoblot analysis**. Cell lysates were prepared in lysis buffer containing 1% Nonidet P-40 and protease inhibitor cocktail (Roche)[46] with or without RNase (1 mg of cell lysate was mixed with 0.4 U/ml RNase A and 16.6 U/ml RNase T1 and incubated at 37 °C for 15 min)[15]. Soluble proteins were immunoprecipitated using anti-Flag (Sigma, F2426, 1:50 dilution), anti-Myc (Sigma, E6654, 1:50 dilution), anti-CREB1 (Cell Signaling Technology, 9197, 1:50 dilution) antibodies, or IgG of the same isotype from the same species as a negative control (Sigma, A2909 or A0919, 1:50 dilution). An aliquot of the total lysate (5%, v/v) was included as a control. Immunoblotting was performed with horseradish peroxidase (HRP)-conjugated anti-Myc (Sigma, SAB4200742, 1:2000 dilution), HRP-conjugated anti-Flag (Sigma, A8592, 1:2000 dilution), HRP-conjugated anti-GFP (Santa Cruz, sc-8334, 1:500 dilution), HRP-conjugated anti-β-actin (Sigma, A3854, 1:5000 dilution), anti-phospho-(Ser/Thr) PKA substrate (Cell Signaling Technology, 9621, 1:500 dilution), anti-VP35 (Creative Diagnostics, CABT-B292 or CABT-B1099, 1:1000 dilution), anti-L (Creative Diagnostics, CABT-B1095, 1:500 dilution), anti-NP (Sino Biological, 40443-MM07, 1:1000 dilution), anti-AKIP1 (Abcam, ab135996, 1:1000 dilution), anti-pCREB1-S133 (Abcam, ab32096, 1:1000 dilution), anti-CREB1 (Cell Signaling Technology, 9197, 1:1000 dilution), anti-Thrombomodulin (Abcam, ab109189, 1:1000 dilution), anti-SerpinB2 (Abcam, ab137588, 1:1000 dilution), anti-PRKACA (BD Biosciences, 610980, 1:1000 dilution), anti-IFN-β (Proteintech, 69013-1-Ig, neutralization 25 ng/ml), anti-VASP (Cell Signaling Technology, 3132, 1:1000 dilution), anti-pVASP-S157 (Cell Signaling Technology, 3111, 1:1000 dilution), anti-pPRKACA-T197 (Cell Signaling Technology, 5661, 1:1000 dilution), anti-Akt (Cell Signaling Technology, 9272, 1:1000 dilution), anti-Akt-T308 (Cell Signaling Technology, 9275, 1:1000 dilution), or anti-Akt-S473 (Cell Signaling Technology, 4060, 1:1000 dilution) antibodies. The antigen–antibody complexes were visualized via chemiluminescence (ECL system, GE Healthcare or Immobilon Western Chemiluminescent HRP Substrate, Millipore). A PageRuler Western marker (Thermo) was used as a molecular weight standard. Full scan blots of immunoblot can be found in the source data file.

**Purification of EBOV VP35 protein**. EBOV VP35 proteins were expressed as His-tagged fusion proteins in *E. coli* BL21 (DE3) in a Luria broth medium. EBOV VP35 protein expression was induced at an $OD_{600}$ (optical density at 600 nm) of 0.6 with 0.2 mM isopropyl-D-thiogalactopyranoside (IPTG) and continued for 4 h at 37 °C. Cells were then harvested and suspended in buffer (20 mM Tris and 8 M urea, pH 8.0). Suspended cells were sonicated and clarified by centrifugation at $13,800 \times g$ at 4 °C for 15 min. Then, VP35-His protein was purified using a Ni column (Ni NTA beads 6FF), eluted with buffer (20 mM Tris, 8 M urea, and 500 mM imidazole, pH 8.0), and dialyzed to buffer (20 mM Tris-HCl and 300 mM NaCl, pH 8.0) for renaturation. The purity of the samples was determined by SDS-PAGE.

**Reverse transcription and quantitative RT-PCR**. Total cellular RNA or viral RNA was prepared using the RNeasy mini (QIAGEN, USA) or viral RNA mini (QIAGEN, USA) kit, respectively, according to the manufacturer's protocol. For cDNA synthesis, 0.5 μg of RNA was first digested with a gDNA eraser to remove contaminated DNA and then reverse transcribed using ReverTra Ace qPCR RT master mix with gDNA remover (FSQ-301, Toyobo) in a 20 μL reaction volume. Then, 1 μL cDNA was used as a template for quantitative PCR. The sequences of the primers used are shown in Supplementary Table 1. The samples were denatured at 95 °C for 2 min, followed by 40 cycles of amplification (15 s at 94 °C for denaturation, 60 s at 60 °C for annealing and extension). Quantitative RT-PCR (qRT-PCR) was performed using SYBR Green Real-time PCR Master Mix (QPK-201, Toyobo) with the QuantStudio 6 Flex multicolor real-time PCR detection system (ABI). Relative mRNA levels were normalized to GAPDH and calculated using the $2^{-\Delta\Delta CT}$ method[47]. Means (upper limit of the box) ± s.e.m. (error bars) of three independent experiments are presented in the figures.

**In situ proximity ligation assay**. The Duolink in situ proximity ligation assay (PLA) (Sigma) was used to detect the endogenous association of AKIP1 and VP35 in cells. In brief, HepG2 cells plated on glass coverslips were transfected with a plasmid expressing GFP-VP35. After fixation with 4% formaldehyde, cells were permeabilized with 0.3% Triton X-100 in PBS for 15 min. After blocking with blocking buffer (Sigma, DUO82007), the cells were incubated with rabbit anti-AKIP1 (Thermo, PA5-66385, 1:100 dilution), rabbit anti-CREB1 (Cell Signaling Technology, 9197, 1:100 dilution) or rabbit anti-L (Creative Diagnostics, CABT-B1095, 1:50 dilution), and mouse anti-VP35 (Creative Diagnostics, CABT-B292, 1:100 dilution) or mouse anti-PRKACA (BD Biosciences, 610980, 1:100 dilution) primary antibodies. Anti-VP35 or anti-AKIP1 alone was employed as a negative control. Nuclei were stained with DAPI (blue). The red fluorescent spots generated from the DNA amplification-based reporter system combined with oligonucleotide-labeled secondary antibodies were detected with a Zeiss LSM 800 Meta confocal microscope (Carl Zeiss, built-in software ZEN2.3). Mean complex numbers (upper limit of the box) ± s.e.m. (error bars) from ten cells are presented.

**Immunofluorescence microscopy**. Cells were transfected, fixed, permeabilized, and blocked as described above. Then, after incubation with anti-AKIP1 (Thermo, PA5-66385, 1:100 dilution), anti-VP35 (Creative Diagnostics, CABT-B292, 1:100 dilution), anti-NP(Sino Biological, 40443-MM07 or 40443-T62, 1:100 dilution), anti-PRKACA (BD Biosciences, 610980, 1:100 dilution), anti-CREB1 (Cell Signaling Technology, 9197, 1:100 dilution), or anti-pCREB1-S133 (Abcam, ab32096, 1:50 dilution) antibodies overnight at 4 °C, the cells were washed three times with blocking solution and then incubated with FITC- or TRITC-conjugated goat anti-rabbit (or anti-mouse) IgG. The cells were then stained with DAPI after washing and imaged using a laser scanning confocal microscope (Zeiss LSM 800 Meta, built-in software ZEN2.3) with a 63× oil immersion lens. The intensity of CREB1 in the cytoplasm and nucleus was analyzed by ImageJ software.

**Gene silencing using siRNA**. For gene knockdown in HepG2 cells, cells maintained in six-well plates were transfected with 100 pmol AKIP1 small interfering RNA (siRNA) (sense, 5′-GCAGUUGAUUCUGGACAAATT-3′; antisense, 5′-UUUGUCCAGAAUCAACUGCTT-3′), 100 pmol CREB1 siRNA (sense, 5′-GGCCUGCAAACAUUAACCATT-3′; antisense: 5′-UGGUUAAUGUUUGCAG GCCTT-3′) or the same concentration of scrambled siRNA (sense, 5′-UUCUCC GAACGUGUCACGUTT-3′; antisense, 5′-ACGUGACACGUUCGGAGAATT-3′) purchased from Genepharma Technologies (Suzhou, China) with Lipofectamine 3000 (Invitrogen) according to the manufacturer's recommendations.

**Generation of the AKIP1 knockout HepG2 cell line using the CRISPR-Cas9 system**. AKIP1 knockout cell lines were generated using the pSpCas9 (BB)-2A-Puro (PX459) (Addgene plasmid no. 48139) vector with a single guide RNA (sgRNA) targeting the human *AKIP1* gene. The sgRNA sequences (target sequence 1: TGGCGGCCGCAGCGCTGAAT; target sequence 2: CATGTCTATCGTTAT-CACAG) were designed using a CRISPR design web tool (http://crispr.mit.edu). The DNA sequences encoding sgRNAs were cloned into the CRISPR-Cas9 vector. Cells were transfected with the sgRNA vectors, and stable clones were screened by puromycin (1 μg/ml). Frameshift mutations in the *AKIP1* gene were confirmed by sequencing (at least 30 T-vector clones were sequenced) and immunoblotting.

**Luciferase reporter assay**. Cells were seeded in 24-well plates and transfected with the indicated amount of Flag-VP35, 200 ng of the CRE reporter plasmid (Promega, USA), and 4 ng of *Renilla* luciferase plasmid. An empty vector was used to ensure the same plasmid concentration in each well. After stimulation with or without 25 μM FSK for 4 h, the cells were harvested, and the luciferase activity of the cell lysates was analyzed with the dual-luciferase reporter assay system (Promega, USA) using a TD-20/20 (or GloMax 20/20) luminometer (Promega, USA). Values were obtained by normalizing the luciferase values to the *Renilla* values. Fold induction was determined by setting the vector transfection without Flag-VP35 and without FSK as a value of 1.

**PKA kinase activity assay**. A PKA kinase activity kit was used (Enzo, USA) for the assay, which is based on a solid phase enzyme-linked immunosorbent assay (ELISA) that utilizes a specific peptide as a substrate for PKA and a polyclonal antibody that recognizes the phosphorylated form of the substrate. Briefly, HepG2 cells were infected with Ad-VP35 or Ad-GFP at an MOI of 10. Forty-eight hours after infection, the cells were lysed, and the protein concentration was determined using the bicinchoninic acid (BCA) method. The PKA kinase activity in HepG2 cells was assayed according to the manufacturer's instructions. In brief, samples were added to wells of the PKA substrate microtiter plate. Then, diluted ATP was added to each well to initiate the reaction and incubated at 30 °C for 90 min. Next, a phosphospecific substrate antibody was added. After incubating at room temperature for 60 min, the diluted anti-rabbit IgG:HRP conjugate was added to each well and incubated at room temperature for 30 min. After washing, TMB substrate was added and incubated at room temperature for 30 min. Finally, a stop solution was added, and the absorbance was measured at 450 nm. The relative kinase activity in cell lysate = (Average absorbance of the sample − Average absorbance of the blank)/Quantity of crude protein used per assay.

**Determination of the intracellular cAMP level**. The intracellular cAMP level was determined by a direct cAMP enzyme immunoassay kit (Enzo, USA). Briefly, WT and $AKIP1^{-/-}$ HepG2 cells were infected with EBOV (MOI = 10) for 72 h. The cells were lysed in 0.1 M HCl and centrifuged at room temperature, and the supernatant was directly subjected to the cAMP assay according to the manufacturer's recommendations.

**EBOV trVLPs assay**. The replication of EBOV in the cells was evaluated by the minigenome system[26]. Briefly, producer cells (p0) were cotransfected with p4cis-vRNA-RLuc (250 ng) and pCAGGS-T7 (250 ng) expressing T7 RNA polymerase and four plasmids expressing EBOV proteins (pCAGGS-NP (125 ng), pCAGGS-VP35 (125 ng), pCAGGS-VP30 (75 ng), and pCAGGS-L (1000 ng)). One day after transfection, the medium was replaced with a medium containing 5% FBS and then incubated for another 3 days. Target cells (p1 or later) were transfected with pCAGGS-NP (125 ng), pCAGGS-VP35 (125 ng), pCAGGS-VP30 (75 ng), pCAGGS-L (1000 ng), and pCAGGS-Tim (250 ng), incubated for 24 h, infected with transcription- and replication-competent virus-like particles (trVLPs) from the p0 (or p1) supernatant for 24 h, and then cultured for another 3 days in medium containing 5% FBS. Viral replication was either determined by intracellular luciferase activities using the Renilla-Glo luciferase assay kit (Promega, E2710) after cell lysis by passive lysis buffer (PLB, Promega) or by viral RNA levels determined by reverse transcription and qRT-PCR.

**Sucrose density gradient centrifugation**. Sucrose gradient centrifugation was employed to fraction cell lysates as previously described[15]. WT and $AKIP1^{-/-}$ HepG2 cells grown in 100-mm dishes were transfected with plasmids expressing pCAGGS-NP (1.25 μg), pCAGGS-VP35 (1.25 μg), pCAGGS-VP30 (0.75 μg), pCAGGS-L (10 μg), pCAGGS-T7 (2.5 μg), and p4cis-vRNA-RLuc (2.5 μg). After 48 h, the cells were harvested and homogenized in 1 ml of lysis buffer (10 mM HEPES (pH 7.5), 12.5% sucrose, 1 mM EDTA, and 1× protease/phosphatase inhibitor cocktail) for 15 min at 4 °C. The samples were then sequentially centrifuged at 700 and $1000 \times g$ for 5 min. Subsequently, the supernatants were layered onto continuous 25 to 60% sucrose gradients containing 10 mM HEPES (pH 7.5) and 1 mM $MgCl_2$ and then centrifuged at $137,000 \times g$ for 2.5 h using an Optima MAX-XP (Beckman). Fractions were collected from top to bottom in 14-drop (300 μL) fractions. Then, the proteins in each fraction were subjected to immunoprecipitation and immunoblot analysis.

**RNA immunoprecipitation (RNA-IP) assays**. WT and *AKIP1*-depleted HepG2 cells were transfected with the plasmids pCAGGS-NP (1.25 μg), pCAGGS-VP35 (1.25 μg), pCAGGS-VP30 (0.75 μg), pCAGGS-L (10 μg), pCAGGS-T7 (2.5 μg), and p4cis-vRNA-RLuc (2.5 μg). After 48 h, the cells were lysed and assayed with the ChIP-IT High Sensitivity kit (53040, Active Motif, USA) as previously described. The CREB1 antibody (Cell Signaling Technology, 9197, 1:25) was used to precipitate potential bound viral RNA. Precipitated RNA was reverse transcribed and quantified using real-time PCR. Primer pairs from EBOV 3′ leader (3Le, 1-469) and 5′ trailer (5Tr, 18283-18959) as described previously were employed[15]. The PCR primers are shown in Supplementary Table 1.

**EBOV infection assay**. WT or *AKIP1*-depleted HepG2 cells grown in 12-well plates were incubated with EBOV Mayinga strain at the indicated MOI determined by virus titration in Vero E6 cells (which was significantly higher than that observed in HepG2 cells, Supplementary Fig. 3a), for viral proliferation assay and microscopy, at 37 °C for 1 h. Then, cells were washed three times with PBS, and a fresh medium was added in the presence/absence of the indicated concentration of H89, T-705, or 666-15 and incubated at 37 °C for 72 h (for microscopy) or 96 h (for viral proliferation assay). Subsequently, cells were fixed with 4% formaldehyde for immunofluorescence microscopy or lysed in 1 ml of TRIzol reagent (Invitrogen) for RNA extraction, as required by the BSL-4 laboratory. The viral RNA was then precipitated by adding 1 ml of 100% ethanol and purified using a QIAamp viral RNA mini kit (QIAGEN, USA) according to the manufacturer's protocol in which the lysis step was omitted. EBOV RNA was quantitatively analyzed by reverse transcription and qRT-PCR. The viral titers were determined by plaque formation assay. Briefly, 10-fold serially diluted samples were added to a 96-well plate containing $1 \times 10^4$ Vero E6 cells per well. Cells were observed for cytopathic effects, and the titers were expressed as the median tissue culture infective dose ($TCID_{50}$). All work with live EBOV was performed in BSL-4 containment.

**mRNA expression microarray analysis**. The SurePrint G3 human gene expression 8x60k v2 microarray from Agilent was used for transcriptome analysis. Briefly, HepG2 cells were transfected with the plasmids Flag-VP35 or Flag. Forty-eight hours later, total RNA was extracted using the RNeasy mini kit (QIAGEN, USA) according to the manufacturer's protocol. After first-strand cDNA synthesis with Poly-dT primer and second-strand cDNA synthesis, the cDNA was subjected to Gene Expression Microarray analysis following the manufacturer's protocol. The chips were scanned with a GeneChip Scanner G2565CA (Agilent) and analyzed with Agilent Feature Extraction (v10.7) and GeneSpring software V13 (Agilent).

**Tail bleeding time**. The bleeding time was measured as previously described in ref. [48]. WT and *Akip1*$^{-/-}$ C57BL/6 N mice (10 weeks, female) were infected with Ad-VP35 or Ad-null ($2 \times 10^9$ PFU) via tail vein injection twice at 24 h intervals. Three days after the second injection, WT mice were administered intraperitoneally with/without 2 mg/kg 666-15 (dissolved in 10% DMSO, 40% PEG300, 5% Tween-80, and 45% saline) three times at 24 h intervals. Six days later, the mice were challenged with LPS (5 mg/kg) or saline via their tail vein. After 4 h, the tails of the mice were cut transversely from the tip at 1 cm with surgical scissors. The bleeding tail stump was immediately placed in normal saline at 37 °C, and the time was measured until the bleeding stopped.

**Coagulation analysis**. Mouse coagulation was measured using a previously described method, with modifications[43]. WT and *Akip1*$^{-/-}$ mice were infected with Ad-VP35 or Ad-null ($3 \times 10^9$ PFU) via tail vein injection twice at 24 h intervals. Six days later, the mice were injected with LPS (5 mg/kg) or saline via their tail vein. After 4 h, blood prothrombin time (PT) and serum FIB concentrations were determined using an automated blood coagulation analyzer (Rayto, RAC-030) according to the manufacturer's instructions.

**Statistical analyses**. Graphical representation and statistical analyses were performed using Prism 8 software (GraphPad Software). Unless indicated otherwise, the results are presented as the means (upper limit of the box) ± s.e.m. (error bars) from three or more independent experiments conducted in duplicate. An unpaired two-tailed *t*-test was used for the analysis of the two groups. Survival curves were analyzed by log-rank test. Data were considered significant when $P < 0.05$ (*), $P < 0.01$ (**), and $P < 0.001$ (***).

**Reporting summary**. Further information on research design is available in the Nature Research Reporting Summary linked to this article.

## Data availability

The mRNA expression microarray data generated in this study have been deposited in the GEO database under the accession code GSE188630. All data were available from the corresponding author upon reasonable request. Source data are provided with this paper.

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

## Acknowledgements

This work was supported by the National Laboratory of Biosafety of China [2019AACP-ZD01] and the National Major Science and Technology Projects of China [2018ZX09711003-005-005]. We thank J. U. Jung (University of Southern California) for providing the cGAS cDNA construct.

## Author contributions

G.X., X.L., and C.C. designed and supervised the study. L.Z., T.G., J.J., Y.L., and H.L. performed the experiments. K.X., G.Z., L.Zhao, R.C., and W.Z. provided the EBOV minigenome system. Y.H., L.Zhang, and Y.W. performed the experiment related to EBOV infection in the BSL-4 laboratory. Y.Hu, P.L., Y.J., W.Y., Q.D., G.W., T.Z., D.W., C.S., Y.B., and X.Z. analyzed the data. X.X. provided experimental guidance. L.Z, X.L., and C.C. wrote the manuscript.

## Competing interests

The authors declare no competing interests.
