## [Peer Review File · Nature Communications]

REVIEWER COMMENTS, first round

Reviewer #1 (Remarks to the Author):

The article „Ebola virus VP35 hijacks PKA-CREB1 pathway for replication and pathogenesis by AKIP1 association” from Zhu et al. describes an interaction of the Ebola virus (EBOV) protein VP35 with the PKA-CREB1 pathway, which they suggest to be essential for viral replication, and speculate to ultimately result in dysregulation of coagulation, thus contributing to EBOV pathogenesis.

Overall, this is an intriguing hypothesis, and the authors provide a substantial amount of data using different experimental setups to support it. Unfortunately, many of the experiments shown miss proper controls or show other issues, and overall I am not convinced that the data provided are sufficient to support the authors’ conclusions. Further, there are a number of issues in the description of methods, which maybe just constitute simple language issues, but which raise additional concerns regarding the reliability of the obtained data. Also, figure 2 is missing completely. Altogether, this raises some significant concerns, and at the very least the issues detailed below need to be adequately addressed prior to publication. Further, the study does not at all address the mechanism how the PKA-CREB1 pathway could be essential for the virus life cycle.

Major comments:

- 1) line 96: The authors observe that the VP35 mutant F239A/K319A/R322A resulted in an abrogation of association with AKIP1, and point out that this mutant is deficient in RNA binding. However, they do not conclusively show that RNA plays a role in the interaction between VP35 and AKIP1. They should repeat their CoIP experiments in presence of RNase in order to clearly demonstrate (or exclude) a role of RNA for this interaction.
- 2) line 106 / figure 1d: Necessary controls to exclude cross-reactivity of secondary antibodies are missing.
- 3) line 106 / figure 1d and e: Given the high MOI (10) and time after infection (48 hrs according to figure legend, 96 hrs according to the methods section), inclusion bodies should be visible throughout the whole cell, and in many more cells. The authors need to provide a convincing explanation why this is not the case (frankly, I can’t think of one).
- 4) line 121 / extended data figure 4b: AKIP1 knock-out has not been thoroughly confirmed, as the sequence shown in extended data figure 4B does not show a bi-allelic knock-out. The authors should confirm knock-out by Western blotting (in the methods they indicated that they have done so, but no data are shown), and provide sequencing data for both alleles.
- 5) line 134 / figure 2: Figure 2 is missing completely (instead, figure 3 is provided twice), and needs to be provided.
- 6) line 155 / figure 3c: As above, the inclusion body staining does not at all reflect what is to be expected based on the experimental parameters. This needs to be explained.
- 7) line 162 / figure 3f: Necessary controls to exclude presence of contaminating DNA, which often can be an issue for RNA-IPs, are missing (-RT controls). Also, in the methods section insufficient experimental details are provided - e.g., did the authors perform a DNase digest?
- 8) line 180 / figure 4: Essential cytotoxicity data are missing.
- 9) line 180 / figure 4: For the luciferase assays, a second control luciferase expressed from an expression plasmid should be used for normalization to control for any effects of the AKIP1 knockout or of the inhibitors on plasmid-driven gene expression. Otherwise, such effects could result in false-positive results.

10) Figure 4g: It seems extremely (!) unlikely that an infection with an MOI of 10 (as indicated in the figure) or 1 (as indicated in the methods section) results in absolutely no progeny virus after 48 hrs, even in presence of strong inhibitors. Further, when looking at extended data figure 6c, also in untreated cells relatively few cells are infected (much less than one would expect at this MOI). Since these results are key for the conclusions of the manuscript, this result should be confirmed by 1) performing a proper growth kinetic (low MOI, infection of 90-100% confluent cells, daily sampling for 5 to 7 days), and 2) by determining the IC50 of 666-15 in infection with EBOV (virus, not trVLPs!).

11) line 243: As the authors correctly state, the mechanism by which CREB1 contributes to the virus life cycle remains completely unclear - this is a major weakness of this manuscript.

12) line 482: For the AKIB1-/- mice only information regarding one allele is provided, and here a 114bp deletion in exon 4 is described. This does not result in a frameshift, but only in an in-frame deletion, and is therefore not a proper KO.

Minor comments:

13) The whole manuscript needs to be checked / corrected by an English native speaker.

14) line 28: According to current nomenclature the term Ebola hemorrhagic fever is outdated and should be replaced by Ebola virus disease (EVD).

15) line 46: The authors' use of nomenclature is wrong - the authors should use current ICTV nomenclature.

16) line 46: The Ebolavirus genus contains six species; Bombali ebolavirus is missing.

17) line 49: The statement regarding case fatality rate is misleading, as current data indicate that the CFR for Ebola virus is rather 40-60%.

18) line 55: VIB as abbreviation is not introduced in the manuscript text, only in the abstract.

19) lines 66-71: Statements here should be referenced.

20) line 92: "reciprocal tags" is confusing - the authors appear to mean "reciprocal pulldown"

21) line 112: "trVLP" is an abbreviation for "transcription and replication-competent virus-like particle"

22) line 117 / figure 1f: The authors claim that VP35 enhances the capability of AKIP1 to bind PRKACA, based on the observation that when precipitating PRKACA the amount of coprecipitated AKIP1 is increased; however, they do not explore the possibility that VP35 could interact with both AKIP1 and PRKACA at the same time, and thus increase the amount of co-precipitated AKIP1 by providing a bridging interaction. This possibility could be explored by including a control in which PRKACA and VP35 are tested in absence of AKIP1.

23) line 155 / figure 3c: Arrows should be shown in all channels, and explained in the legend.

24) line 170-171: What do the authors mean with "modified trVLP"? The authors should explain how this trVLP system differs from the original trVLP system, and they should indicate the source of the trVLP systems they used.

25) line 202: instead of "biomarker of death" authors should use "biomarker for fatal outcome".

26) line 433 / figure 4a: The authors state that vRNA levels are shown, but the graph shows luciferase levels.

27) line 434 / figure 4c/d: Why do the authors assess activity in p1/p2 in some experiments, and

p0/p1 in others?

28) line 510: When an isotype control antibody is used or the IP, the authors have to make sure that they use the same type of antibodies as used for the IP. As the described antibodies are either mouse IgG or rabbit IgG, the isotype controls have to be mouse IgG and rabbit IgG as well and not anti-mouse IgG. Otherwise, the IPs in Figure 3e and Extended data figure 1b lack proper negative controls.

29) line 548: Please provide details regarding the used blocking buffer (composition or commercial source).

30) line 607: The use of term "minigenome assay" here is misleading; the assay is properly called "trVLP assay".

31) line 651: Why were the cells lysed in TRIzol, but the RNA was extracted with the QIAamp viral RNA mini kit? This will most likely not work / makes no sense.

32) Extended data figure 3: The purpose of this figure is unclear to me.

33) Extended data figure 7b: The labels in this panel are too small to read.

34) Extended data figure 8 is not mentioned in the results section, but only in the discussion. These results should already be mentioned in the results section.

Reviewer #2 (Remarks to the Author):

The authors identified a novel association between AKIP1, a PKA-interacting protein, and VP35. They show that EBOV VP35 significantly potentiates the activation of the PKA-CREB1 signaling pathway. This has two distinct consequences: first, AKIP1 and activated CREB1 are found in VIBs. Pharmacological inhibition of both, PKA and CREB1, or AKIP1 depletion inhibits viral replication. Secondly, CREB1-regulated coagulation-related genes are upregulated by VP35, which is abrogated by PKA or CREB1 inhibitor or AKIP1 depletion. This could have major implications in Ebola-infected patients as VP35 may cause coagulation disorder. Mice infected with Ad-VP35 demonstrated higher mortality upon LPS challenge than AKIP1^{-/-} mice or mice treated with CREB1 inhibitor. This is a novel, interesting study shedding light not only on EBOV replication and potential novel antivirals but also on the role of VP35 in coagulation disorder. However, further evidence would be required to strengthen the conclusions.

Major points:

- 1) Figure 2 is missing! Experiments related to Fig. 2 cannot be reviewed.
- 2) The authors observe that interaction between AKIP1 and VP35 is abrogated upon mutating AAs that are responsible for binding dsRNA (shown in overexpression experiments). Are cellular RNAs involved in the binding? The authors need to exclude that viral RNA might enhance the binding in infection experiments – they could use the trVLP system including mutated VP35 to proof that viral RNA does not play a role in this interaction.
- 3) The main point of the manuscript, enhanced activation of PRK by VP35, enhanced association to AKIP1 and enhanced downstream activation, particularly the CREB1-regulated genes, needs to be properly controlled through a mutant VP35 abrogating the interaction to AKIP1 in order to show specificity.
- 4) The authors demonstrate that a high proportion of activated phosphorylated CREB1 is recruited to the VIB? Is there a direct association with any of the viral proteins within the VIBs, e.g. VP35? What is the mechanism of nuclear activated CREB1 to be recruited to IBs?
- 5) An important mechanistic question remains: what is the exact mechanism of inhibition of viral replication? What is the function of CREB1 in the VIBs? KO or KD of CREB1 could shed light on this question. Why is CREB1 recruited to the VIBs?

Minor points:

Many spelling mistakes, manuscript text needs to be carefully revised.

Line 45ff: According to Kuhn et al. 2019, all subcategories of diseases caused by Ebolaviruses fall under the abbreviation EBOD, whereas EVD is specifically used for Zaire EBOV. Six species are known by now.

Reviewer #3 (Remarks to the Author):

In this work, authors reveal human A kinase interacting protein (AKIP1) as binding partner of Ebola virus VP35 protein and provide functional insights of this interaction in context to Ebola virus infection. AKIP1, a nuclear protein is a binding partner of catalytic subunit of Protein Kinase A (PRKACA) and facilitates PRKACA nuclear translocation. In this work authors define the role of Ebola virus VP35 protein-AKIP1 interaction in activating PKA-CREB1 signaling pathway and studies its impact on virus replication. Authors utilizes different molecular and imaging techniques to characterize this interaction and mapping the region on each protein for interaction, and further study the effect on virus replication using inhibitors of PKA or CREB1 or knockout cell models of AKIP1 in infection model and trVLP system. The impact of AKIP1 depletion on Ebola virus infection and replication is reasonable, however role PKA-CREB1 axis is less clear specifically of PKA. Overall, these experiments support a model for role of AKIP1 in Ebola virus replication that is quite unique and interesting but impact of PKA-CREB axis and connection to coagulation genes is abit puzzling.

Some of the major concerns are highlighted below that need clarification and additional experiments.

1. Authors identified AKIP1 as interacting partner of VP35 through yeast two hybrid screening, why the data from the original screen is included. Was AKIP1 the only protein identified? It is also missing the methods for yeast two hybrid screening.
2. Authors mapped the interaction of AKIP1 to C-terminal region of VP35. Interestingly, they identified RNA binding residues are important for this interaction, this would suggest that this interaction could be mediated by dsRNA. Figure 1a, authors perform coimmunoprecipitation (co-IP) in HepG2 cells but there is no mention of depletion of dsRNA, so its not clear how authors came to conclusion in in lines 99-100, that VP35-AKIP1 interaction occurs in absence of dsRNA. This co-IP needs to be performed in presence or absence of dsRNA or in vitro co-IP with purified proteins may help to address this question.
3. Is this interaction conserved among VP35 of different members of Filovirus genera?
4. As AKIP1 interacted with C-terminus of VP35 protein, a region critical for innate immune antagonism, does this interaction impact VP35 function in suppression of interferon responses?
5. In Fig. 1f, authors claim that AKIP1 binding to PRKACA is enhanced in presence of VP35 however co-IP was performed with PRKACA pull down, it raises the question whether VP35 interacts with PRKACA as well. Authors need to include co-IP PRKACA and VP35 in presence and absence AKIP1 to clarify the interaction of VP35 and PRKACA.
6. Furthermore, authors claim that VP35 enhances PKA activation (Figure 1h and i), but data presented is not sufficient to establish if its due to VP35 interaction with AKIP1. The data in Fig. 1i, is hard to interpret as overall PKA activation is abrogated in the absence of AKIP1. To evaluate this, Fig.1 h and i, could include VP35 F239/K319/R322A mutant as control (shows loss of interaction with AKIP1) and co-IPs can be performed in absence of AKIP1 (preferably with AKIP1 knockdown). One of the hallmark of PKA activation is the phosphorylation of PKA on T197, authors should evaluate phosphorylation at this site (pmid:12372837, pmid:19293561).
7. If AKIP1 is important for virus replication, does this interaction impact VP35-L, VP35-L interaction in the replication complex?
8. Figure 3 is duplicated in multiple figures, as in Fig 2 and 5. The original data for Fig 2 and 5 are missing.
9. One of the consequences of PKA activation is suppression of innate immune responses during RNA virus infection (<https://doi.org/10.1371/journal.ppat.1006648>). This is particularly interesting given the role of VP35 in interferon antagonism. If VP35 indeed potentiates PKA activation, will it also impact activation of innate immune responses. In the virus action studies, this could be evaluated by assessing gene expression of IFN and ISGs.

10. In Fig 4a, authors show that depletion of AKIP1 leads to decrease in trVLP activity, does overexpression of AKIP1 will enhance this activity?
11. The accumulation of cellular cAMP is associated with activation of PKA, does Ebola infection triggers increase in cellular cAMP levels? This should be evaluated in presence and absence of AKIP1.
12. It is interesting to note that nuclear proteins, AKIP1 and CREB1 get recruited in viral inclusion bodies, so in this scenario how AKIP1 maintains its association with PKA or AKIP1 interaction is enhanced in presence of VP35? Does PKA also get recruited to inclusion bodies? In Figure 1g, authors show that VP35 enhances nuclear translocation of PRKACA, so how does this fits with interaction data? Can authors include nuclear/cytoplasmic assessment as performed for CREB1 in Fig. 3. Based on the proposed model in Extended Fig. 9 authors suggest that PKA is recruited into replication machinery, can authors provide data for this function?
13. Authors perform detailed characterization to implicate role of AKIP1 and PKA-CREB1 axis in virus replication, however PKA has important role in controlling innate immune responses which is not addressed in this work. As VP35 plays versatile roles in virus replication and innate immune antagonism, it is imperative to clarify relevance of this interactions on VP35 functions during virus infection.
14. CREB1 data in Fig. 3 and 4 is the most interesting finding of this work. The inhibition of CREB1 shows a very drastic decrease in trVLP activity and virus RNA levels which is dependent on presence of AKIP1. CREB1 is downstream target of many other kinases with PKA being one of them, however PKA inhibition or activation did not show substantial effect on virus replication (Fig 4c), this would suggest that PKA is not the only kinase involved in this process. To implicate PKA-CREB1 axis involvement, can authors evaluate recruitment of CREB1 in viral inclusion bodies in absence of PKA?
15. In addition, so it will be important to establish effect of other downstream substrates of PKA on Ebola virus replication and substantiate the relevance of AKIP1 mediated PKA signaling pathway.
16. Microarray analysis with vector or VP35 does not provide relevant information. This would be of much interest if authors performed this experiment with VP35 in presence or absence of AKIP1 as well in Ebola infection. These data would directly implicate pathways that are regulated by AKIP1 and are important to infection.
17. In Fig. 7, there many pathways that were enriched in microarray analysis, but authors focused on coagulation specific pathway and did not even discuss the consequence of other pathways. CREB1 could be regulated by multiple other kinases, one of the enriched pathways in Fig 7b is P13K-AKT signaling pathways, interestingly AKT is also known activator of CREB1, so why authors did not evaluate relevance of this pathway to Ebola virus infection? CREB1 indeed seems to have robust impact on virus but kinases responsible for CREB1 could be more than more in context of virus infection.
18. Authors did not include the cell viability data with inhibitors. This data will be important to assess the antiviral effects are not due to cell toxicity.
19. Authors should include internal loading control for coIPs and Western blot, Fig. 1a-c, 1f, 3d-e.
20. One minor comment is that authors should also specify how many independent experiments were performed especially for co-Ips, western blots and imaging.

REVIEWER COMMENTS

Reviewer #1 (Remarks to the Author):

Overall, this is an intriguing hypothesis, and the authors provide a substantial amount of data using different experimental setups to support it. Unfortunately, many of the experiments shown miss proper controls or show other issues, and overall I am not convinced that the data provided are sufficient to support the authors' conclusions. Further, there are a number of issues in the description of methods, which maybe just constitute simple language issues, but which raise additional concerns regarding the reliability of the obtained data. Also, figure 2 is missing completely. Altogether, this raises some significant concerns, and at the very least the issues detailed below need to be adequately addressed prior to publication. Further, the study does not at all address the mechanism how the PKA-CREB1 pathway could be essential for the virus life cycle.

We apologize for the missing data due to a careless mistake during online submission. We have carefully checked the revision to ensure that the correct figures were uploaded. The revised manuscript has been substantially improved according to the reviewer's suggestions. We hope all the concerns have been addressed adequately.

Major comments:

1) line 96: The authors observe that the VP35 mutant F239A/K319A/R322A resulted in an abrogation of association with AKIP1, and point out that this mutant is deficient in RNA binding. However, they do not conclusively show that RNA plays a role in the interaction between VP35 and AKIP1. They should repeat their CoIP experiments in presence of RNase in order to clearly demonstrate (or exclude) a role of RNA for this interaction.

Response: The Co-IP experiment was repeated in the presence/absence of RNase (the mixture of RNase A and RNase T1), and the results showed that the interaction between VP35 and AKIP1 does not depend on the presence of RNA (new Fig. 1e, lines 105-106).

2) line 106 / figure 1d: Necessary controls to exclude cross-reactivity of secondary antibodies are missing.

Response: Controls with the secondary antibody only (without anti-AKIP1 primary antibody) have been added to the revised manuscript to exclude the cross-reactivity of secondary antibodies (new Fig. 2a, lines 119-122).

3) line 106 / figure 1d and e: Given the high MOI (10) and time after infection (48 hrs according to figure legend, 96 hrs according to the methods section), inclusion bodies should be visible throughout the whole cell, and in many more cells. The authors need to provide a convincing explanation why this is not the case (frankly, I can't think of one).

Response: The TCID₅₀ of live EBOV in this study was titered in Vero E6 cells, and the infection efficiency in the HepG2 cells used in this study was substantially lower than that in Vero E6 cells (non-interferon-producing cells) (new Extended Data Fig. 3a, lines 119-122, 756-758). The MOI=10 was calculated based on the viral titer in infected Vero E6 and HepG2 cell numbers. Therefore, viral inclusion bodies (VIBs) were not visible throughout the whole cell or in many more cells. We have carefully addressed the concerns in the revised Methods section and in the description of the related results. We have also repeated the experiments and the results are presented in the new Fig. 2a (lines 119-122).

4) line 121 / extended data figure 4b: AKIP1 knock-out has not been thoroughly confirmed, as the sequence shown in extended data figure 4B does not show a bi-allelic knock-out. The authors should confirm knock-out by Western blotting (in the methods they indicated that they have done so, but no data are shown), and provide sequencing data for both alleles.

Response: CRISPR-mediated AKIP1 knockout in HepG2 cells was carefully verified by T-vector sequencing. At least 30 T-vector clones were sequenced, all of which produced a single nucleotide peak with only little baseline noise at the mutation site. Double and multiple peaks were not observed at the mutation site, indicating that all gene copies had been edited successfully. The related experiment has been described in detail in the Methods section (lines 682-684). The confirmation of AKIP1 knockout by Western blot is shown in the new Fig. 4e.

5) line 134 / figure 2: Figure 2 is missing completely (instead, figure 3 is provided twice), and needs to be provided.

Response: We apologize for the missing data, and Fig. 2 (new Fig. 4) has been provided in the revised manuscript.

6) line 155 / figure 3c: As above, the inclusion body staining does not at all reflect what is to be expected based on the experimental parameters. This needs to be explained.

Response: We have addressed the concern in the response to major point 3. New data with cells containing typical inclusion bodies are provided in the new Fig. 5c, d (lines 173-174).

7) line 162 / figure 3f: Necessary controls to exclude presence of contaminating DNA, which often can be an issue for RNA-IPs, are missing (-RT controls). Also, in the methods section insufficient experimental details are provided - e.g., did the authors perform a DNase digest?

Response: As suggested by the reviewer, the RNA was reverse transcribed using ReverTra Ace qPCR RT master mix with gDNA remover (FSQ-301). The detailed experimental procedure has been described in the Methods section of the revised manuscript (lines 633-636).

8) line 180 / figure 4: Essential cytotoxicity data are missing.

Response: According to the reviewer's suggestion, the cytotoxicity of the drugs used in this study was tested by performing the CCK-8 assay and trypan blue staining. No chemicals, except the CREB1 inhibitor KG-501, showed significant cytotoxicity at the concentration tested. Data have been provided in the new Extended Data Fig. 8e (lines 201-205, 208-210). KG-501-related data were removed from the revised manuscript.

9) line 180 / figure 4: For the luciferase assays, a second control luciferase expressed from an expression plasmid should be used for normalization to control for any effects of the AKIP1 knockout or of the inhibitors on plasmid-driven gene expression. Otherwise, such effects could result in false-positive results.

Response: As suggested by the reviewer, we assessed the potential effect of *AKIP1*^{-/-} and drug treatment on plasmid-driven gene expression using a firefly luciferase expressing plasmid (pGL3-Promoter, VT1726). No significant variation in firefly luciferase expression was observed (new Extended Data Fig. 8a), indicating that plasmid-driven gene expression was unlikely to be affected in the samples shown in Fig. 4 (new Fig. 6).

10) Figure 4g: It seems extremely (!) unlikely that an infection with an MOI of 10 (as indicated in the figure) or 1 (as indicated in the methods section) results in absolutely no progeny virus after 48 hrs, even in presence of strong inhibitors. Further, when looking at extended data figure 6c, also in untreated cells relatively few cells are infected (much less than one would expect at this MOI). Since these results are key for the conclusions of the manuscript, this result should be confirmed by 1) performing a proper growth kinetic (low MOI, infection of 90-100% confluent cells, daily sampling for 5 to 7 days), and 2) by determining the IC50 of 666-15 in infection with EBOV (virus, not trVLPs!).

Response: The virus infection experiment in Fig. 4g (new Fig. 6h) was performed at an MOI based on the titer in Vero E6 cells (as a response to major point 3). As suggested by the reviewer, we performed the following experiments: 1) HepG2 cells were infected at a low MOI (MOI=0.1), and the cell supernatant was collected daily for 5 days after infection (more than 50% of cells were infected)

to determine the virus titer (new Fig. 6g). 2) HepG2 cells were infected at a low MOI (MOI=0.1) and treated with different concentrations of 666-15 (0, 0.1, 0.2, 0.4, 0.8 and 1.6 μ M) for 4 days. Subsequently, the IC50 of 666-15 that inhibited the proliferation of EBOV was determined (blue line in new Fig. 6e, lines 217-220).

11) line 243: As the authors correctly state, the mechanism by which CREB1 contributes to the virus life cycle remains completely unclear - this is a major weakness of this manuscript.

Response: We have tried to clarify the mechanism by providing a large amount of new data in the revision. 1) VP35/NP/L and CREB1 were observed in the same complex (new Fig. 5e, f); 2) CREB1 can enrich 3' vRNA of EBOV, the initiation site for virus replication (new Fig. 5g); 3) VP35-L association and VIBs formation were significantly potentiated by AKIP1 and active CREB1, based on the comparison of wild type cells with AKIP1 knockout, CREB1 knockdown, or 666-15-treated cells (new Extended Data Fig. 7). These data collectively supported the hypothesis that the AKIP1-PKA-CREB1 pathway was involved in EBOV replication by increasing the interaction of viral RNA-dependent RNA polymerase (RdRp) with its cofactor VP35, although other mechanisms could not be excluded.

12) line 482: For the AKIB1^{-/-} mice only information regarding one allele is provided, and here a 114bp deletion in exon 4 is described. This does not result in a frameshift, but only in an in-frame deletion, and is therefore not a proper KO.

Response: We apologize that the wrong sequencing data (from another *Akip1*^{-/-} mouse) were provided by the company generating the mice (Cyagen Bioscience). The knockout mice used in this study had a deletion of 339 bp, of which the fourth exon was deleted by 28 bp (lines 568-569). The sequencing data for the mice we used were provided by Cyagen Bioscience Inc., as follows:

Sequencing Results:

Positive animals

Mouse ID: 57, 58, 61, 62 (Deleted 339 bp)

GCACCAGGAAAGTAAACACAGAATTGTTGATCTGACTCCA--del 339 bp--GAGGAGCAACCCATGTCTACCGTTATCACAGAAGGAAACC

Minor comments:

13) The whole manuscript needs to be checked / corrected by an English native speaker.

Response: The language of this manuscript has been improved by the American Journal Experts English language editing service, and typos and errors have been corrected throughout the revised manuscript.

14) line 28: According to current nomenclature the term Ebola hemorrhagic fever is outdated and should be replaced by Ebola virus disease (EVD).

Response: "Ebola hemorrhagic fever" has been replaced with "Ebola virus disease (EVD)" in the revised manuscript (lines 30-31).

15) line 46: The authors' use of nomenclature is wrong - the authors should use current ICTV nomenclature.

Response: The current ICTV nomenclature was used in the revision (lines 49-53).

16) line 46: The Ebolavirus genus contains six species; Bombali ebolavirus is missing.

Response: *Bombali ebolavirus* has been added to the revised manuscript (line 53).

17) line 49: The statement regarding case fatality rate is misleading, as current data indicate that the CFR for Ebola virus is rather 40-60%.

Response: The average EVD case fatality rate has been changed to 40-60% based on the current data (line 54).

18) line 55: VIB as abbreviation is not introduced in the manuscript text, only in the abstract.

Response: The definition for the abbreviation VIBs has been introduced in the revised manuscript (line 125).

19) lines 66-71: Statements here should be referenced.

Response: The reference has been added to the revised manuscript (lines 69-72).

20) line 92: “reciprocal tags” is confusing - the authors appear to mean “reciprocal pulldown”

Response: The sentence was corrected as suggested (lines 96-97).

21) line 112: “trVLP” is an abbreviation for “transcription and replication-competent virus-like particle”

Response: “Replication and transcription-competent virus-like particle” has been changed to “transcription and replication-competent virus-like particle” in the revised manuscript (line 119).

22) line 117 / figure 1f: The authors claim that VP35 enhances the capability of AKIP1 to bind PRKACA, based on the observation that when precipitating PRKACA the amount of coprecipitated AKIP1 is increased; however, they do not explore the possibility that VP35 could interact with both AKIP1 and PRKACA at the same time, and thus increase the amount of co-precipitated AKIP1 by providing a bridging interaction. This possibility could be explored by including a control in which PRKACA and VP35 are tested in absence of AKIP1.

Response: The Co-IP results showed that PRKACA failed to associate with VP35 in the absence of AKIP1 (new Extended Data Fig. 2e, f), suggesting that VP35 does not interact with PRKACA directly (lines 113-117).

23) line 155 / figure 3c: Arrows should be shown in all channels, and explained in the legend.

Response: Arrows have been added to all channels and have been annotated in the legend of the new Fig. 5c (lines 499-500).

24) line 170-171: What do the authors mean with “modified trVLP”? The authors should explain how this trVLP system differs from the original trVLP system, and they should indicate the source of the trVLP systems they used.

Response: “Modified trVLP” here refers to the trVLP carrying the luciferase reporter gene. It is the same system as reported by Hoenen *et al.* in 2014 (PMID: 25285674) (line 119). We replaced the misleading description with “trVLP” in the revised manuscript (lines 118-119).

25) line 202: instead of “biomarker of death” authors should use “biomarker for fatal outcome”.

Response: Changes were made in the revised manuscript (line 302).

26) line 433 / figure 4a: The authors state that vRNA levels are shown, but the graph shows luciferase levels.

Response: Fig. 4a (new Fig. 6a) shows luciferase levels, and Fig. 4b (new Fig. 6b) shows vRNA levels.

27) line 434 / figure 4c/d: Why do the authors assess activity in p1/p2 in some experiments, and p0/p1 in others?

Response: In the p0 of trVLP system, viral replication was initiated by T7-driven transcription of viral RNA, and thus the luciferase level only partially reflects the viral replication in p0. trVLP from p1 might reflect viral life cycle better. Therefore, the results from p0 and p1 were included in the revised manuscript (new Fig. 6c, lines 201-205).

28) line 510: When an isotype control antibody is used or the IP, the authors have to make sure that they use the same type of antibodies as used for the IP. As the described antibodies are either mouse IgG or rabbit IgG, the isotype controls have to be mouse IgG and rabbit IgG as well and not anti-mouse IgG. Otherwise, the IPs in Figure 3e and Extended data figure 1b lack proper negative controls.

Response: A clear statement of “IgG of same isotype from the same species was used as negative control” was provided in the revised manuscript (lines 604-605).

29) line 548: Please provide details regarding the used blocking buffer (composition or commercial source).

Response: The blocking buffer was purchased from Sigma (DUO82007) (line 649).

30) line 607: The use of term “minigenome assay” here is misleading; the assay is properly called “trVLP assay”.

Response: “EBOV minigenome assay” has been changed to “EBOV trVLP assay” in the revised manuscript (line 717).

31) line 651: Why were the cells lysed in TRIzol, but the RNA was extracted with the QIAmp viral RNA mini kit? This will most likely not work / makes no sense.

Response: According to the SOP of the BSL-4 laboratory in Wuhan, EBOV-containing samples used for RNA extraction must be completely inactivated by TRIzol (containing phenol, thiocyanic acid compound with guanidine) for safe handling before they are taken out from the BSL-4 laboratory. Then, ethanol-precipitated RNA was further purified using QIAmp viral RNA mini kit, in which the lysis step with AVL cell lysis buffer (containing guanidine thiocyanate) was omitted (line 762-766). The compatibility of the method was confirmed using the minigenome system.

32) Extended data figure 3: The purpose of this figure is unclear to me.

Response: The transmission electron microscopy assay was intended to show that the EBOV minigenome system used in this study normally produces VIBs and trVLP. These data have been deleted from the revised manuscript.

33) Extended data figure 7b: The labels in this panel are too small to read.

Response: Clear labels were provided in the panel (new Extended Data Fig. 9b).

34) Extended data figure 8 is not mentioned in the results section, but only in the discussion. These results should already be mentioned in the results section.

Response: Extended Data Fig. 8 (new Extended Data Fig. 4d) has been moved to the Results section of the revised manuscript (lines 141-143).

Reviewer #2 (Remarks to the Author):

The authors identified a novel association between AKIP1, a PKA-interacting protein, and VP35. They show that EBOV VP35 significantly potentiates the activation of the PKA-CREB1 signaling pathway. This has two distinct consequences: first, AKIP1 and activated CREB1 are found in VIBs. Pharmacological inhibition of both, PKA and CREB1, or AKIP1 depletion inhibits viral replication. Secondly, CREB1-regulated coagulation-related genes are upregulated by VP35, which is abrogated by PKA or CREB1

inhibitor or AKIP1 depletion. This could have major implications in Ebola-infected patients as VP35 may cause coagulation disorder. Mice infected with Ad-VP35 demonstrated higher mortality upon LPS challenge than AKIP1^{-/-} mice or mice treated with CREB1 inhibitor. This is a novel, interesting study shedding light not only on EBOV replication and potential novel antivirals but also on the role of VP35 in coagulation disorder. However, further evidence would be required to strengthen the conclusions.

Major points:

1) Figure 2 is missing! Experiments related to Fig. 2 cannot be reviewed.

Response: We apologize for the missing data, and Fig. 2 (new Fig. 4) has been provided in the revised manuscript.

2) The authors observe that interaction between AKIP1 and VP35 is abrogated upon mutating AAs that are responsible for binding dsRNA (shown in overexpression experiments). Are cellular RNAs involved in the binding? The authors need to exclude that viral RNA might enhance the binding in infection experiments – they could use the trVLP system including mutated VP35 to proof that viral RNA does not play a role in this interaction.

Response: The Co-IP experiment was repeated in the presence/absence of RNase (the mixture of RNase A and RNase T1), and the results showed that the interaction between VP35 and AKIP1 does not depend on the presence of RNA (new Fig. 1e, lines 105-106).

3) The main point of the manuscript, enhanced activation of PRK by VP35, enhanced association to AKIP1 and enhanced downstream activation, particularly the CREB1-regulated genes, needs to be properly controlled through a mutant VP35 abrogating the interaction to AKIP1 in order to show specificity.

Response: According to the reviewer's suggestion, the VP35 mutant (KFR/AAA) was included as a control, and it was not observed to potentiate PKA activity compared to wild-type VP35 (new Fig. 3c and Extended Data Fig. 4c, lines 132-135, 137-139).

4) The authors demonstrate that a high proportion of activated phosphorylated CREB1 is recruited to the VIB? Is there a direct association with any of the viral proteins within the VIBs, e.g. VP35? What is the mechanism of nuclear activated CREB1 to be recruited to IBs?

Response: Upon viral infection, a portion (not a high portion) of phosphorylated CREB1 was recruited to VIBs, as shown by immunofluorescence staining and the sucrose density gradient analysis. The *in situ* association between CREB1 and VP35 in the VIBs was determined using the PLA assay (new Extended Data Fig. 6a-c). Since VP35 binds to AKIP1 and then recruits PKA in the cytoplasm, we speculated that PKA phosphorylates a portion of CREB1 in the cytoplasm and recruited it into VIBs, in addition to its phosphorylation in the nucleus. A discussion of this possibility was provided in the revised manuscript.

5) An important mechanistic question remains: what is the exact mechanism of inhibition of viral replication? What is the function of CREB1 in the VIBs? KO or KD of CREB1 could shed light on this question. Why is CREB1 recruited to the VIBs?

Response: We have tried to clarify the mechanism by adding a large amount of new data in the revision. 1) VP35/NP/L and CREB1 were observed in the same complex (new Fig. 5e, f); 2) CREB1 can enrich 3' vRNA of EBOV, the initiation site for virus replication (new Fig. 5g); and 3) VP35-L association and VIBs formation were significantly potentiated by AKIP1 and active CREB1 based on the comparison of wild type cells with AKIP1 knockout cells, CREB1 knockdown, or 666-15 treated cells (new Extended Data Fig. 7). These data collectively supported the hypothesis that the AKIP1-PKA-CREB1 pathway was involved in EBOV replication by increasing the interaction of viral RNA-dependent RNA polymerase (RdRp) with its cofactor VP35, although other mechanisms could not be excluded.

Minor points:

Many spelling mistakes, manuscript text needs to be carefully revised.

Response: The language of this manuscript has been improved by the American Journal Experts English language editing service, and typos and errors have been corrected throughout the revised manuscript.

Line 45ff: According to Kuhn et al. 2019, all subcategories of diseases caused by Ebolaviruses fall under the abbreviation EBOD, whereas EVD is specifically used for Zaire EBOV. Six species are known by now.

Response: The manuscript was revised according to the comments (lines 49-53).

Reviewer #3 (Remarks to the Author):

Some of the major concerns are highlighted below that need clarification and additional experiments.

1. Authors identified AKIP1 as interacting partner of VP35 through yeast two hybrid screening, why the data from the original screen is included. Was AKIP1 the only protein identified? It is also missing the methods for yeast two hybrid screening.

Response: By performing yeast two-hybrid screening, several proteins that potentially interact with VP35 were identified but have not yet been confirmed by detailed investigations. We would be happy to provide the primary data, if required. The yeast two-hybrid screening method has been added to the revised manuscript (lines 587-598). Changes were made in the revised manuscript.

2. Authors mapped the interaction of AKIP1 to C-terminal region of VP35. Interestingly, they identified RNA binding residues are important for this interaction, this would suggest that this interaction could be mediated by dsRNA. Figure 1a, authors perform coimmunoprecipitation (co-IP) in HepG2 cells but there is no mention of depletion of dsRNA, so its not clear how authors came to conclusion in in lines 99-100, that VP35-AKIP1 interaction occurs in absence of dsRNA. This co-IP needs to be performed in presence or absence of dsRNA or in vitro co-IP with purified proteins may help to address this question.

Response: The Co-IP experiment was repeated in the presence/absence of RNase (the mixture of RNase A and RNase T1), and the results showed that the interaction between VP35 and AKIP1 does not depend on the presence of RNA (new Fig. 1e, lines 105-106).

3. Is this interaction conserved among VP35 of different members of Filovirus genera?

Response: The interaction of AKIP1 with RESTV (nonpathogenic in humans) VP35 filovirus was investigated as suggested. Human AKIP1 interacts with VP35 of EBOV but not with VP35 of human nonpathogenic RESTV (new Fig. 1c, lines 97-98).

4. As AKIP1 interacted with C-terminus of VP35 protein, a region critical for innate immune antagonism, does this interaction impact VP35 function in suppression of interferon responses?

Response: The depletion of AKIP1 inhibited the transcription of IFN- β and IFN-stimulated gene 15 (ISG15), but VP35 still suppressed IFN- β and ISG15 expression in *AKIP1* knockout cells to a similar extent as in wild-type cells. Based on these data, the interaction of AKIP1 with VP35 at the C terminus did not significantly alter the function of VP35 in suppressing interferon responses (new Extended Data Fig. 4e, f, lines 143-149).

5. In Fig. 1f, authors claim that AKIP1 binding to PRKACA is enhanced in presence of VP35 however co-IP was performed with PRKACA pull down, it raises the question whether VP35 interacts with PRKACA as well. Authors need to include co-IP PRKACA and VP35 in presence and absence AKIP1 to clarify the interaction of VP35 and PRKACA.

Response: The Co-IP results showed that PRKACA failed to associate with VP35 in the absence of AKIP1 (new Extended Data Fig. 2e, f), suggesting that VP35 does not interact with PRKACA directly (lines 113-117).

6. Furthermore, authors claim that VP35 enhances PKA activation (Figure 1h and i), but data presented is not sufficient to establish if its due to VP35 interaction with AKIP1. The data in Fig. 1i, is hard to interpret as overall PKA activation is abrogated in the absence of AKIP1. To evaluate this, Fig. 1h and i, could include VP35 F239/K319/R322A mutant as control (shows loss of interaction with AKIP1) and co-IPs can be performed in absence of AKIP1 (preferably with AKIP1 knockdown). One of the hallmark of PKA activation is the phosphorylation of PKA on T197, authors should evaluate phosphorylation at this site (pmid:12372837, pmid:19293561).

Response: According to the reviewer's suggestion, the VP35 mutant (KFR/AAA) was included as a control, and it was not observed to obviously potentiate PKA activity compared to wild-type VP35 (new Fig. 3c and Extended Data Fig. 4c, lines 132-135, 137-139). Additionally, the Co-IP results showed that PRKACA failed to associate with VP35 in the absence of AKIP1 (new Extended Data Fig. 2e, f), suggesting that VP35 does not activate PRKACA directly (lines 113-117).

As suggested by the reviewer, we detected the phosphorylation of PKA at T197 induced by EBOV trVLPs, and the results showed that EBOV trVLPs moderately increased the phosphorylation of at PKA T197 in HepG2 cells (new Extended Data Fig. 4b, lines 135-136).

7. If AKIP1 is important for virus replication, does this interaction impact VP35-L, VP35-L interaction in the replication complex?

Response: Duolink PLA experiments were performed in the presence and absence of AKIP1. AKIP1 expression significantly potentiates the VP35-L interaction in VIBs (new Extended Data Fig. 7, lines 187-189).

8. Figure 3 is duplicated in multiple figures, as in Fig 2 and 5. The original data for Fig 2 and 5 are missing.

Response: We apologize for the missing data, and Fig. 2 (new Fig. 4) has been provided in the revised manuscript.

9. One of the consequences of PKA activation is suppression of innate immune responses during RNA virus infection (<https://doi.org/10.1371/journal.ppat.1006648>). This is particularly interesting given the role of VP35 in interferon antagonism. If VP35 indeed potentiates PKA activation, will it also impact activation of innate immune responses. In the virus action studies, this could be evaluated by assessing gene expression of IFN and ISGs.

Response: The expression of IFN and ISG was assessed in the revision. As expected, VP35 expression inhibited the transcription of IFN- β and IFN-stimulated gene 15 (ISG15) in an AKIP1-independent manner upon SeV stimulation. However, unexpectedly, AKIP1 knockout suppressed the transcription of IFN- β and ISG15, at least in the cells we tested (new Extended Data Fig. 4e, f, lines 143-149).

10. In Fig 4a, authors show that depletion of AKIP1 leads to decrease in trVLP activity, does overexpression of AKIP1 will enhance this activity?

Response: Our results showed that AKIP1 overexpression did not enhance trVLP activity. In contrast, excess AKIP1 results in a moderately inhibited trVLP activity, which might result from the impaired PRKACA-CREB1 recruitment due to the altered AKIP1, PRKACA and VP35 molecular ratio in the cells (new Extended Data Fig. 8d, lines 199-201).

11. The accumulation of cellular cAMP is associated with activation of PKA, does Ebola infection triggers increase in cellular cAMP levels? This should be evaluated in presence and absence of AKIP1.

Response: Intracellular cAMP concentrations were significantly increased by EBOV infection in wild-type HepG2 cells and to a lesser extent in AKIP1-depleted HepG2 cells (new Fig. 3e, line 139-141, 711-716).

12. It is interesting to note that nuclear proteins, AKIP1 and CREB1 get recruited in viral inclusion bodies, so in this scenario how AKIP1 maintains its association with PKA or AKIP1 interaction is enhanced in presence of VP35? Does PKA also get recruited to inclusion bodies? In Figure 1g, authors show that VP35 enhances nuclear translocation of PRKACA, so how does this fits with

interaction data? Can authors include nuclear/cytoplasmic assessment as performed for CREB1 in Fig. 3. Based on the proposed model in Extended Fig. 9 authors suggest that PKA is recruited into replication machinery, can authors provide data for this function?

Response: 1) Immunofluorescence assays confirmed that VP35 (NP) and CREB1 colocalized in VIBs, and sucrose density gradient and immunoprecipitation analyses showed that VP35 (NP) and CREB1 existed in the same fraction.

2) PKA and VIBs exhibited significant colocalization, indicating that PKA is also recruited into VIBs (new Extended Data Fig. 5a, lines 164-168).

3) The VP35-AKIP1 association not only activates PKA and promotes PRKACA nuclear translocation, to induce CREB1 phosphorylation in the nucleus (new Fig. 3b and Fig. 4b, c), but also forms a complex with AKIP1-PKA-CREB1 in VIBs to promote EBOV replication (new Fig. 5).

4) According to the reviewer's suggestion, we performed the cytoplasmic/nuclear assessment and the data are presented in new Fig. 5c, d (lines 173-174).

13. Authors perform detailed characterization to implicate role of AKIP1 and PKA-CREB1 axis in virus replication, however PKA has important role in controlling innate immune responses which is not addressed in this work. As VP35 plays versatile roles in virus replication and innate immune antagonism, it is imperative to clarify relevance of this interactions on VP35 functions during virus infection.

Response: As suggested by the reviewer, several experiments were conducted to clarify the relevance of the concern. 1) VP35 expression inhibited the transcription of IFN- β and ISG15 in an AKIP1-independent manner upon SeV stimulation, which suggested that the interaction of AKIP1 with VP35 at the C terminus did not significantly affect the innate immune antagonistic activity of VP35 (new Extended Data Fig. 4e, f, lines 143-149). 2) H89 still significantly inhibited EBOV trVLPs replication in the presence of the JAK1 inhibitor (ruxolitinib) and anti-IFN- β antibody, which excluded the possibility that PKA promoted the viral replication mainly by attenuating the innate antiviral responses (new Extended Data Fig. 8f, lines 205-208).

14. CREB1 data in Fig. 3 and 4 is the most interesting finding of this work. The inhibition of CREB1 shows a very drastic decrease in trVLP activity and virus RNA levels which is dependent on presence of AKIP1. CREB1 is downstream target of many other kinases with PKA being one of them, however PKA inhibition or activation did not show substantial effect on virus replication (Fig 4c), this would suggest that PKA is not the only kinase involved in this process. To implicate PKA-CREB1 axis involvement, can authors evaluate recruitment of CREB1 in viral inclusion bodies in absence of PKA?

Response: As suggested by the reviewer, we evaluated the effect of the PKA inhibitor H89 on the recruitment of CREB1 into VIBs by performing immunofluorescence staining. The formation of VIBs was significantly impaired in the absence of PKA, and no detectable CREB1 was observed in shrunken VIBs (new Fig. 5a, lines 168-170).

15. In addition, so it will be important to establish effect of other downstream substrates of PKA on Ebola virus replication and substantiate the relevance of AKIP1 mediated PKA signaling pathway.

Response: Indeed, we found that VP35 participates in viral replication through the AKIP1-PKA-CREB1 axis. Additionally, viral VP35-mediated PKA activation resulted in the phosphorylation of a number of PKA substrates, such as CREB1 presented in this work and vasodilator-stimulated phosphoprotein (VASP) (new Extended Data Fig. 4d), a well-defined PKA substrate involved in platelet activation and aggregation. However, we could not exclude the possibility that other downstream substrates of PKA are involved in EBOV replication through AKIP1. In addition, Flag-VP35-transfected HepG2 cells exhibited the considerably increased phosphorylation of a number of PKA substrates compared with Flag-vectors, as evidenced by anti-pPKA substrate immunoblotting, indicating that these unidentified PKA substrates might be associated with the function of EBOV VP35 and are worthy of in-depth investigations in the future (new Fig. 3c).

16. Microarray analysis with vector or VP35 does not provide relevant information. This would be of much interest if authors

performed this experiment with VP35 in presence or absence of AKIP1 as well in Ebola infection. These data would directly implicate pathways that are regulated by AKIP1 and are important to infection.

Response: The microarray analysis was intended to screen VP35-regulated genes, and the involvement of AKIP1/PKA was further assessed by performing qRT-PCR in wild-type/*AKIP1*-depleted cells, as well as the cells treated with/without PKA inhibitors (new Fig. 7c, lines 239-243). We have performed a microarray analysis of EBOV-infected HepG2 cells with or without AKIP1, and significant differences in the enriched Gene Ontology (GO) terms were observed in the two cell lines upon EBOV infection; further studies are still undergoing. Because that the data are not directly related to the effect of VP35, we hope to publish the results in the future.

17. In Fig. 7, there many pathways that were enriched in microarray analysis, but authors focused on coagulation specific pathway and did not even discuss the consequence of other pathways. CREB1 could be regulated by multiple other kinases, one of the enriched pathways in Fig 7b is P13K-AKT signaling pathways, interestingly AKT is also known activator of CREB1, so why authors did not evaluate relevance of this pathway to Ebola virus infection? CREB1 indeed seems to have robust impact on virus but kinases responsible for CREB1 could be more than more in context of virus infection.

Response: As suggested by the reviewer, we investigated the effect of VP35 on the phosphorylation of Akt at S473 and T308 using Western blotting. VP35 did not increase the phosphorylation of Akt at S473 and T308 (new Extended Data Fig. 10, lines 260-264). Thus, the PI3K-Akt signaling pathway may not be responsible for regulating virus replication through the PI3K-Akt signaling pathway.

18. Authors did not include the cell viability data with inhibitors. This data will be important to assess the antiviral effects are not due to cell toxicity.

Response: According to the reviewer's suggestion, the cytotoxicity of the drugs used in this study was tested by performing the CCK-8 assay and trypan blue staining. Except for KG-501, the other drugs tested did not show significant cytotoxicity at the concentration used in the assay. KG-501-related data were removed and the new data have been provided as the new Extended Data Fig. 8e (lines 201-205, 208-210).

19. Authors should include internal loading control for coIPs and Western blot, Fig. 1a-c, 1f, 3d-e.

Response: Immunoblotting of β -actin was included as an internal loading control in the above figures.

20. One minor comment is that authors should also specify how many independent experiments were performed especially for co-IPs, western blots and imaging.

Response: The figure legends were revised according to the reviewer's suggestion.

REVIEWER COMMENTS, second round

Reviewer #1 (Remarks to the Author):

This is a revised version of a previously submitted manuscript. The authors have carefully addressed previously raised concerns, and show a significant amount of new data, that greatly improve the manuscript. However, one major concern remains that should be addressed:

Major points:

line 197 / extended data figure 8B: AKIP1-depletion results in significantly reduced VP35 expression in IFA (extended Fig. 8B), but not in Western Blotting (Figure 4D). How do the authors explain this? These IFA data are also in contradiction to the response to my comment #9, where the authors indicate that AKIP1-knockout does not affect plasmid-driven gene expression.

Minor points:

line 44/50: The authors use two different abbreviations (EBOD and EVD) for the same term. EVD is the commonly used one.

line 59: The abbreviation VIB needs to be defined in the text at its first occurrence (currently it is introduced in line 125).

Reviewer #2 (Remarks to the Author):

The authors made a significant effort to address most of the issues. However, it is still unclear whether RNA plays a role in the interaction between VP35 and AKIP1. This was pointed out by all three reviewers, particularly as it is so striking to see the mutations in VP35 that abrogate binding to AKIP1 are known to bind dsRNA. The presented experiment is not convincing as there is no control on whether the RNase treatment was successful and therefore the negative result (RNA is not involved) has no meaning. To my point of view, a thorough analysis on this point will be necessary, in CO-IP or in vitro experiments and also in trVLP-infected cells before this manuscript should be considered for publication.

Point-to-Point Responses to Reviewer's Comments

REVIEWER COMMENTS

Reviewer #1 (Remarks to the Author):

This is a revised version of a previously submitted manuscript. The authors have carefully addressed previously raised concerns, and show a significant amount of new data, that greatly improve the manuscript. However, one major concern remains that should be addressed:

Major points:

line 197 / extended data figure 8B: AKIP1-depletion results in significantly reduced VP35 expression in IFA (extended Fig. 8B), but not in Western Blotting (Figure 4D). How do the authors explain this? These IFA data are also in contradiction to the response to my comment #9, where author's indicate that AKIP1-knockout does not affect plasmid-driven gene expression.

Response: Fig.4D showed the VP35 level in wild-type and *AKIP1* knockdown cells infected with the non-replicative adenovirus bearing VP35 genes, while Extended Data Fig. 8b showed the VP35 level in wild-type and *AKIP1* depleted cells infected with the Luc-trVLPs, in which trVLP replication was significantly inhibited by *AKIP1* depletion. Upon trVLP infection, most of VP35 and NP were recruited into the VIBs compartment other than in cytosol upon trVLPs infection (Fig. 5a, Extended Data Fig. 5c, 6a and 7). In accordance with viral RNA level, VP35 and NP levels in the VIBs were significantly downregulated by *AKIP1* depletion, CREB1 knockdown or 666-15 treatment (Fig. 5a, Extended Data Fig. 5c, 6a and 7), which suggested that VP35 and NP level in VIBs may be used as a marker for virus replication since it might be much more stable than free VP35 in the cytoplasm. As control, in the cells infected with non-replicative Ad-VP35, or transfected with Flag-VP35 plasmid (Extended Data Fig. 1c), *AKIP1* showed little if any effect on VP35 expression in *AKIP1* knockdown cells (Fig. 4d), or even potentiated VP35 expression in *AKIP1* knockout cells (Fig. 4e). These results further demonstrated that *AKIP1* regulated viral replication through VP35 induced PKA-CREB1 activation but not plasmid driven expression of VP35. Discussion was made in the revision (line 282-293).

Minor points:

line 44/50: The authors use two different abbreviations (EBOD and EVD) for the same term. EVD is the commonly used one.

Response: As suggested by Dr. Jens H Kuhn et al. (New filovirus disease classification and nomenclature, Nature Review Microbiology, 17: 261-263, 2019) and reviewer #2, disease caused by any virus of *Ebolavirus* genus were named as EBOD, whereas the disease caused by Zaire EBOV was named as EVD.

line 59: The abbreviation VIB needs to be defined in the text at its first occurrence (currently it is introduced in line 125).

Response: Changes were made in the revised manuscript (line 60).

Reviewer #2 (Remarks to the Author):

The authors made a significant effort to address most of the issues. However, it is still unclear whether RNA plays a role in the interaction between VP35 and AKIP1. This was pointed out by all three reviewers, particularly as it is so striking to see the mutations in VP35 that abrogate binding to AKIP1 are known to bind dsRNA. The presented experiment is not convincing as there is no control on whether the RNase treatment was successful and therefore the negative result (RNA is not involved) has no meaning. To my point of view, a thorough analysis on this point will be necessary, in CO-IP or in vitro experiments and also in trVLP-infected cells before this manuscript should be considered for publication.

Response: Thanks for the comments. A Co-IP of G3BP1 and cGAS, whose interaction had been demonstrated to be RNA-dependent (PMID: 31772125), was included as a positive control to show RNase treatment was successful (New Extended Data Fig. 1e, lines 108-110). And, as suggested, similar VP35:AKIP1 association was also demonstrated in the presence/absence of RNA, in the lysates of trVLPs-infected cells (New Extended Data Fig. 1f, lines 110-112).

REVIEWER COMMENTS, third round

Reviewer #1 (Remarks to the Author):

This is a second revision of a previously submitted manuscript. The reviewers have now also addressed my last remaining major comment. I still have one minor suggestion to the authors (see below) regarding taxonomy (and the authors use thereof), but this is something that could be addressed at the galley proof stage.

Minor point:

I still believe that the use of EBOD vs. EVD (and thus talking about ebolaviruses and Ebola viruses (i.e. about viruses at the genus and the species level)) is incredibly confusing for readers not intimately familiar with filovirus taxonomy, and particular for the rather broad readership of Nature communications. Importantly, this distinction between EBOD and EVD also is completely irrelevant for this paper; indeed, EBOD is mentioned only once, and could easily be omitted without impacting the manuscript. Further, abbreviations such as EBOV, SUDV etc. are used for viruses in a species, not the species itself. Finally, if the authors use EVD in a sense that it only covers disease caused by Ebola virus, the word “major” in line 55 is misleading.

Thus, in order to avoid this confusion, I would recommend to change the sentence starting in line 50 by deleting “As the pathogens causing Ebola disease (EBOD),” and further modifying it, so that the sentence should read “The Ebolavirus genus includes six species: Zaire ebolavirus (with the virus Ebola virus (EBOV)), Sudan ebolavirus (Sudan virus (SUDV)), Bundibugyo ebolavirus (Bundibugyo virus, BDBV), Tai Forest ebolavirus (Taï Forest virus (TAFV)), Reston ebolavirus (Reston virus (RESTV), nonpathogenic in humans), and the newly described Bombali ebolavirus (Bombali virus (BOMV)), of which Zaire ebolavirus is the most virulent, with a case fatality rate of 40-60%, and is the cause of EVD outbreaks.”

Reviewer #2 (Remarks to the Author):

The authors addressed all my comments and I am satisfied with their responses.

REVIEWER COMMENTS

Reviewer #1 (Remarks to the Author):

This is a second revision of a previously submitted manuscript. The reviewers have now also addressed my last remaining major comment. I still have one minor suggestion to the authors (see below) regarding taxonomy (and the authors use thereof), but this is something that could be addressed at the galley proof stage.

Minor point:

I still believe that the use of EBOD vs. EVD (and thus talking about ebolaviruses and Ebola viruses (i.e. about viruses at the genus and the species level)) is incredibly confusing for readers not intimately familiar with filovirus taxonomy, and particular for the rather broad readership of Nature communications. Importantly, this distinction between EBOD and EVD also is completely irrelevant for this paper; indeed, EBOD is mentioned only once, and could easily be omitted without impacting the manuscript. Further, abbreviations such as EBOV, SUDV etc. are used for viruses in a species, not the species itself. Finally, if the authors use EVD in a sense that it only covers disease caused by Ebola virus, the word “major” in line 55 is misleading.

Thus, in order to avoid this confusion, I would recommend to change the sentence starting in line 50 by deleting “As the pathogens causing Ebola disease (EBOD),” and further modifying it, so that the sentence should read “The Ebolavirus genus includes six species: Zaire ebolavirus (with the virus Ebola virus (EBOV)), Sudan ebolavirus (Sudan virus (SUDV)), Bundibugyo ebolavirus (Bundibugyo virus, BDBV), Tai Forest ebolavirus (Tai Forest virus (TAFV)), Reston ebolavirus (Reston virus (RESTV), nonpathogenic in humans), and the newly described Bombali ebolavirus (Bombali virus (BOMV)), of which Zaire ebolavirus is the most virulent, with a case fatality rate of 40-60%, and is the cause of EVD outbreaks.”

Response: Changes were made in the revised manuscript (lines 49-55).